# Retrograde trafficking of Argonaute 2 acts as a rate-limiting step for de novo miRNP formation on endoplasmic reticulum–attached polysomes in mammalian cells

Mainak Bose*, Susanta Chatterjee*, Yogaditya Chakrabarty* , Bahnisikha Barman* , Suvendra N Bhattacharyya

**microRNAs are short regulatory RNAs in metazoan cells. Regulation of miRNA activity and abundance is evident in human cells where availability of target messages can influence miRNA biogenesis by augmenting the Dicer1-dependent processing of precursors to mature microRNAs. Requirement of subcellular compartmentalization of Ago2, the key component of miRNA repression machineries, for the controlled biogenesis of miRNPs is reported here. The process predominantly happens on the polysomes attached with the endoplasmic reticulum for which the subcellular Ago2 trafficking is found to be essential. Mitochondrial tethering of endoplasmic reticulum and its interaction with endosomes controls Ago2 availability. In cells with depolarized mitochondria, miRNA biogenesis gets impaired, which results in lowering of de novo–formed mature miRNA levels and accumulation of miRNA-free Ago2 on endosomes that fails to interact with Dicer1 and to traffic back to endoplasmic reticulum for de novo miRNA loading. Thus, mitochondria by sensing the cellular context regulates Ago2 trafficking at the subcellular level, which acts as a rate-limiting step in miRNA biogenesis process in mammalian cells.**

## Introduction

miRNAs constitute an extensive class of small regulatory RNAs that are ~22 nucleotides long and control the expression of more than half of protein-coding genes in humans (1). miRNAs are endogenously transcribed from their respective genes as pri-miRNAs that are processed inside the nucleus by the microprocessor (Drosha-DGCR8) complex to generate a ~60–70-nt pre-miRNA (2, 3, 4). The pre-miRNAs mature in the cytoplasm where they are processed by the RNaseIII endonuclease Dicer to mature miRNAs (5, 6). Mature miRNAs form complex with effector Argonaute proteins to form miRNPs that usually bind to 3′-UTRs of target mRNAs having imperfect complementarities to the respective miRNAs. Binding of miRNAs induces translation repression that is usually accompanied by exonucleolytic degradation of target messages (7).

Because miRNAs are implicated in numerous cellular and developmental pathways (8), it is not surprising that their biogenesis, activity, and turnover are under stringent regulation. Corresponding target mRNAs can also act as a key player in regulating miRNA biogenesis and stability (9, 10, 11, 12, 13, 14). It has been reported of late that a target mRNA induces increased activity of Ago-associated Dicer1 to enhance biogenesis of their cognate miRNAs (15). Target-driven miRNA biogenesis is ensured by increased processivity of the enzyme Dicer1 that in the presence of cognate mRNAs go through higher number of successful Ago-loading cycles for respective miRNAs. This feedback-like mechanism aids in prompt stress recovery of hepatic cells by accelerating the final step of miRNA biogenesis. However, information on the exact subcellular sites of target-driven biogenesis and its regulation has remained unidentified.

Recent evidences have gradually started to unravel the subcellular sites of miRNP assembly and function. The rER has been confirmed as the central nucleation site of miRNA assembly and their interaction with target mRNAs (16, 17). Localization kinetics of a miRNA-targeted message has revealed that a newly formed target mRNA localizes to the ER-attached polysomes first, followed by miRNP binding and onset of translation repression (18). Therefore, it would be interesting to investigate whether rER could act as the site of target-driven miRNA biogenesis. Given so, it would be also fascinating to outline the exact sequence of molecular events that

RNA Biology Research Laboratory, Molecular Genetics Division, Council of Scientific and Industrial Research-Indian Institute of Chemical Biology, Kolkata, India

Correspondence: suvendra@iicb.res.in
Mainak Bose's present address is European Molecular Biology Laboratory, Heidelberg, Germany
Yogaditya Chakrabarty's present address is California Institute of Technology, Pasadena, CA, USA
Bahnisikha Barman's present address is Vanderbilt University School of Medicine, Nashville, TN, USA
*Mainak Bose, Susanta Chatterjee, Yogaditya Chakrabarty, and Bahnisikha Barman contributed equally to this work

occur on the rER membrane in the context of miRNA–target message interaction and miRNA biogenesis.

Although miRNA-mediated translation repression sets on the rER membranes, eventual deadenylation, decapping, and mRNA degradation processes do not occur on the rER (19). The repressed mRNAs are shuttled to early and late endosomes (EEs and LEs) and then to multivesicular bodies (MVBs) where they get uncoupled from the bound miRNPs and undergo degradation. The free miRNP may recycle back to the rER for fresh rounds of repression. However, information on the dynamics of miRNP recycling and its regulation by intracellular organelles is limited.

Mitochondria are extremely dynamic organelles that differ in size and organization depending on the cell type or physiological state. They undergo continuous cycles of fission and fusion, which are apparently two opposite processes which operate in equilibrium to maintain the mitochondrial functional and structural homeostasis in mammalian cells (20). Two human proteins Mfn1 and Mfn2, the homologs of *Drosophila* fzo (fuzzy onions protein), play important role in maintaining the functional and structural aspects of mitochondria in human cells. It has recently been shown how the mitochondrial tethering with rER, a process impaired in cells with depolarized mitochondria, can control the miRNP recycling in human cells (21).

Another group of proteins, mitochondrial uncoupler proteins (UCPs), plays an important role in controlling mitochondrial membrane potential. This protein by uncoupling the electron transport chain can reduce the mitochondrial membrane potential. According to a previous report, defective ER–endosome association due to mitochondria depolarization can cause an increase in miRNPs levels in mammalian cells (22).

In the work described here, we have identified how the accumulation of miRNA-free Ago2, on the LE/MVB membranes in cells defective for mitochondrial tethering with ER, reduces its MVB to ER shuttling to affect de novo miRNP biogenesis on the rER–attached polysome. These findings explore rER as the site of miRNP biogenesis and have shown how the biogenesis process is dependent on the energy content of the cell.

# Results

### De novo–synthesized miRNAs get enriched on rER-attached polysome in mammalian cells

To explore the subcellular sites of miRNP assembly and function, we treated HEK 293 cells ectopically expressing liver-specific miR-122, with the detergent digitonin and measured the activity of miRISC-122 present in digitonin-soluble cytosolic and insoluble membrane fractions (18, 19). As HEK 293 cells do not express miR-122, ectopic expression of miR-122 allow us to follow the miRNA biogenesis process there. Digitonin selectively dissolves the high cholesterol–containing cell membrane, whereas low cholesterol–containing organelle membranes remain intact and can be separated as an insoluble pellet enriched with "membrane fraction" from the soluble "cytosolic fraction" after digitonin treatment. The pellet fractions contain different membranous structures, including rER. To understand the

compartmentalization of miRNAs with different organelles, and more precisely with rER, we did selective isolation of rER in the form of microsome that is devoid of other cellular organelles. Significantly higher specific activity of miRISC-122 was observed in the insoluble membrane fraction compared with the whole cell lysate or the digitonin-solubilized cytosolic fraction (Fig 1A and B). Therefore, functional miRNPs are relatively less abundant in cytosolic fraction, whereas found enriched in the membranous fraction of mammalian cells. To explore the exact subcellular structures with which active miRNPs are attached, we fractionated HEK 293 isotonic cell lysate on a 3–30% iodixanol (OptiPrep) density gradient to separate out individual organelles based on their densities. Endosomes and MVB-rich fractions showed enrichment of both Ago2 and miR-122, although substantial amounts of Ago2 were also detected in ER/polysome fractions (Fig 1C and D). Interestingly, Dicer1 was found to be predominantly present in the ER and polysome fractions. We performed an RISC-mediated target cleavage activity assay in vitro with each fraction and observed a higher specific activity of miRISC-122 present in ER-enriched fractions and polysome. The pre-miR-122 was also almost exclusively found enriched in the fractions positive for ER marker (Fig 1D).

Evidences from earlier reports suggest that the outer surface of the rER acts as the central nucleation site of miRNP assembly and interaction with target mRNAs (16, 17). Earlier, miRNAs and Ago2 were found to be associated with actively translating polysome in HeLa cells (23, 24, 25). A recent report has shown how the de novo–formed mRNAs first localize to ER-bound polysome, before its miRNP binding and translation repression. Microsome, primarily representing rER, was isolated from HEK 293 cells to reconfirm the rER association of cellular miRNPs (18). In isolated microsome, as expected, we detected significant amount of Ago2 protein and enrichment of miRNAs (Fig 1E and F).

How the miRNPs remain attached with the rER membrane? They might attach via mRNAs that are a part of the ribosomal pool present on the rER or may have direct contact sites on the membrane to adhere to the rER. To gain an insight into the possible mode of attachment of miRNPs to the ER membrane, the microsomes were treated with KCl and puromycin to preferentially extract the membrane-attached translating pool of ribosomes from the ER membrane (18) (Fig 1G). Whereas no notable Ago2 or miRNA extraction was observed with no KCl–puromycin treatment, most of Ago2 was found to be almost exclusively get extracted with KCl and puromycin (Fig 1I and J). This suggests most of the rER-attached Ago2 is attached with ribosomal pool and largely absent in the non-ribosomal membrane fraction. Dicer1 showed association with both soluble and residual membrane fractions upon KCl–puromycin treatment. The de novo–synthesized miR-122, accumulated from an inducible expression construct in HEK 293 cells, along with its target mRNA RL-3xbulge-miR-122 was predominantly co-isolated in the supernatant after KCl–puromycin treatment of rER (Fig 1H). Previous reports from our laboratory have also documented how the newly formed mRNAs are trafficked to polysome attached with rER that follows with miRNP binding of target mRNA and miRNA-mediated repression on rER membrane (18). Taken together, these observations corroborate the previous findings and suggest preferential association of both active miRNPs and target mRNAs with the rER-attached polysome.

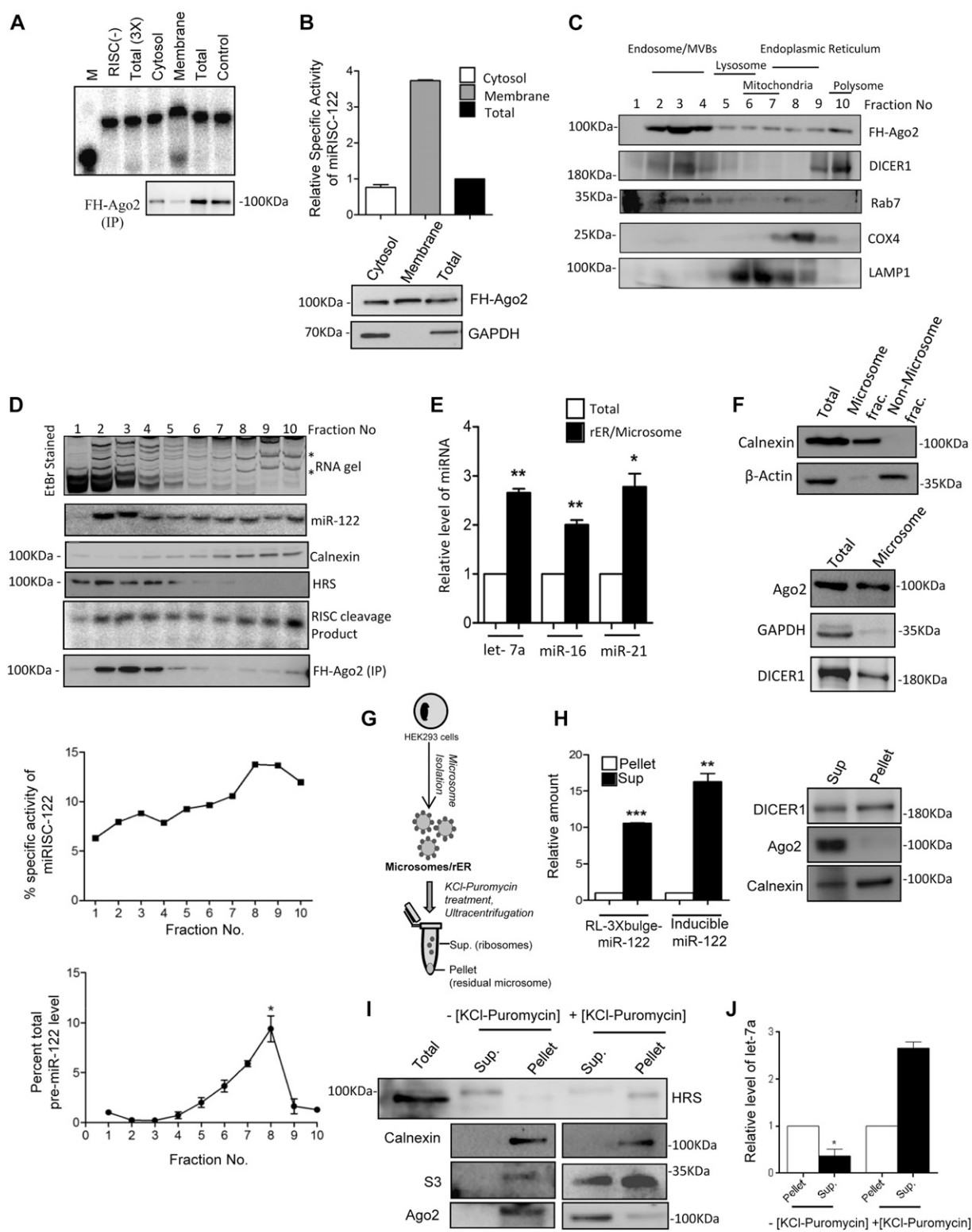

**Figure 1. Attachment of de novo–formed miRNAs with ribosomes attached with the rER membrane.**
**(A, B)** Active miRNPs are membrane associated. FH-Ago2 was immunoprecipitated (IP) from total cell lysate as well as digitonin-soluble cytosolic and insoluble membrane fractions after fractionation of cells performed by treating with 50 µg/ml of digitonin. The IPed materials obtained from cell equivalent amount of each fraction were used in the in vitro RISC cleavage assays. Densitometric quantification of the cleaved RNA band was used for calculating specific activity of miR-122 and normalized against the amount of Ago2 IP following procedures described in the Materials and Methods section and plotted. Representative autoradiogram data obtained from one experimental set have been shown in panel (A) along with the Western blot data for IP FH-Ago2. The relative amount of Ago2 present in each reaction is

## Target-dependent miRNA biogenesis happens on rER

Target-dependent biogenesis of corresponding miRNAs due to enhanced processivity of Dicer1 in the presence of target messages has been reported earlier in mammalian cells (15). However, questions related to the exact subcellular location of the de novo–synthesized miRNPs are unresolved. Because miRNP assembly and interaction with target mRNA is compartmentalized on the surface of the rER, it is plausible that the target-driven de novo–formed miRNPs too get accumulated on the rER. To understand that, we fractionated HEK 293 cells by digitonin treatment as above and the distribution of miRNA in detergent soluble cytosolic and insoluble membrane fractions were analyzed in the presence and absence of target mRNA RL-3xbulge-miR-122 (Fig 2A and B). As per expectation, it was observed that in the steady state, miR-122 primarily accumulates in the membrane-enriched fractions. Also, the observed miR-122 compartmentalization to the membrane fractions was found to be enhanced in the presence of the target mRNA (Fig 2B). The specific activity of miRISC measured in the presence of the target mRNA was much higher in the membrane-associated pool (Fig 2C). The specific activity of miR-122 RISC associated with the rER/microsomes got enhanced in the presence of miR-122 target mRNAs (Fig 2D).

These results were observed in steady-state cells. It would have been interesting to track the fate or subcellular destination of a de novo–formed mature miRNA in the presence of enhanced targets. To follow the kinetics of miRNP generation, we induced pre-miR-122 transcription driven from a doxycyclin-inducible promoter for 24 h in Tet-On HEK 293 cells with doxycyclin and followed subcellular localization of newly formed mature miR-122 after separating cellular organelles on an iodixanol (OptiPrepR) gradient. This helped us not only to track the levels of newly formed miRNAs but also to follow its subcellular distribution. We observed a robust increase in de novo synthesized miR-122 in the rER-enriched fraction, whereas the steady-state level of control miRNA let-7a (non-induced miRNA control) showed less attachment with rER (Fig 2E).

## Kinetics of compartmentalization of miRNAs synthesized in the presence of target mRNAs

The time kinetics of de novo formation of mature miR-122 in the presence of its substrate mRNA was analyzed. The de novo–synthesized miR-122 level was quantified in the presence of its target RL-3xbulge-miR-122. After 24 h of induction, a sharp increase in miR-122 level was detected in microsomal fraction when cells expressing the target reporter mRNA was compared with cells expressing a control mRNA without miRNA-binding sites (Fig 2F). Notably, the microsomal enrichment of de novo–formed miRNAs was higher than the increase observed in total cellular content, signifying that target-driven miRNA enrichment happens preferably on the rER. Having confirmation of enrichment of newly synthesized miRNAs being localized on the rER, the exact sequence of molecular events was probed in subsequent assays. After 4 h of miR-122 induction, a steep increase in cellular mature miR-122 content was noted up to 24 h (Figs 2G and S1A). miR-122 association with rER was analyzed over time, and it was observed that de novo–synthesized mature miR-122 gradually accumulates on the rER and polysomes against time (Figs 2H and S1B). Next, the Ago2 association of de novo–formed miRNAs on rER membranes was analyzed. Ago2 was immunoprecipitated from microsomal fractions, and its miR-122 association was measured against time. It was observed that Ago2 association of de novo–formed miR-122 enhanced specifically for Ago2 present on the rER but not with cytosolic Ago2 (Fig 2I). As expected, Ago2 association of miR-122 showed concomitant increase in its repressive activity as detected by the luciferase assay (Fig 2J).

Therefore, newly formed mature miRNAs localize primarily with rER or ER-bound polysomes in the presence of target mRNAs, and these miRNAs are Ago2 bound and capable of repressing targets. Technical obstacles had prevented us to identify the sequence of event happening in vivo further. Hence, to confirm that the miRNA biogenesis occurs on the rER membrane–bound polysome, we resorted to an in vitro assay system that could closely emulate miRNP formation on membranes in the presence of target mRNA. An in vitro translation reaction was performed with rabbit reticulocyte

---

shown in the Western blots showing levels of FH-Ago2 and GAPDH in panel (B). The FH-Ago2 detected with anti-HA antibody. RISC(-) is the reaction done without any addition of RISC. In Total (3×) reaction, threefold excesss of cytosolic fractions was used. **(C, D)** Subcellular distribution of miRNP components in human cells. FH-Ago2 stable HEK 293 cells transiently expressing miR-122 were lysed under isotonic conditions and subjected to ultracentrifugation on a 3–30% iodixanol gradient for separation of subcellular organelles. Hepatocyte growth actor-regulated tyrosine kinase substrate (HRS), calnexin, LAMP1, COX4, and Rab7 were used as markers of endosomes/multivesicular bodies (MVBs), rER, lysosomes, mitochondria, and late endosomes, respectively. Subcellular distribution of Dicer1 has been observed. FH-Ago2 was IPed from individual fractions using FLAG-specific antibody, and IPed materials were used for in vitro RISC cleavage assay. Specific activity of miR-122 RISC in each fraction has been plotted by normalizing the RISC activity present against the amount of Ago2 precipitated from each fraction. Pre-miR-122 levels in individual fractions have been quantified by qRT-PCR. Values for each fraction have been plotted as percentage of total miR-122 calculated by summing miR-122 present in all the fractions. Positions of ribosomal RNA bands are marked by * in the ethidium bromide–stained gel shown here. **(E, F)** Mature miRNAs are rER-enriched in HEK 293 cells. Relative enrichment of miR-16, miR-21, and let-7a as determined in isolated microsome by qRT-PCR when compared with equal amount of RNA from total cell lysate. Presence of Ago2 and DICER1 in microsome fractions was determined by Western blot. Absence of GAPDH (cytosolic marker) and β-Actin and presence of calnexin (rER marker) in the microsomal fraction confirmed the purity of isolated microsome as compared with protein equivalent amount of non-microsomal or total cell extract. **(G)** Schematic presentation of isolation of polysomes using KCl–puromycin based on HEK 293 cells. Microsomes isolated from HEK 293 cells were treated with 500 mM KCl and 1 mM puromycin before ultracentrifugation at 100,000*g* for 1 h to separate the ribosomal and non-ribosomal pool. **(H)** miRNP and target mRNAs are associated with the rER-attached ribosomes. Western blot data show the distribution of Ago2 in the soluble and pellet fractions after treatment and centrifugations. Calnexin was used as the ER marker protein. Mature miR-122 as well as its target mRNA RL-3xbulge-miR-122 distribution in the abovementioned pellet and soluble fractions were determined by qRT-PCR. **(I)** Exclusiveness of the Ago2 and miRNA extraction with KCl–puromycin. Treated and non-treated extracts were analyzed by Western blots using cell equivalent amounts for each fraction. ER (calnexin), ribosome (S3), and endosomes/MVBs (HRS) marker protein distributions were checked to observe the specificity and purity of the isolation method. Western blot data of HRS protein suggest that minimum contamination of endosomal fractions in the microsome that can account for the observed Ago2 that was found to be extracted with ribosome upon extraction with KCl and puromycin. **(J)** Relative amount of let-7a miRNA associated with soluble and pellet fractions with or without KCl–puromycin treatment was measured. Paired two-tailed *t* tests were used for all comparisons. *P* < 0.05 (*); *P* < 0.01 (**); *P* < 0.001 (***). In (B, D, E, H, J), values are means ± SEM from at least three biological replicates.
Source data are available for this figure.

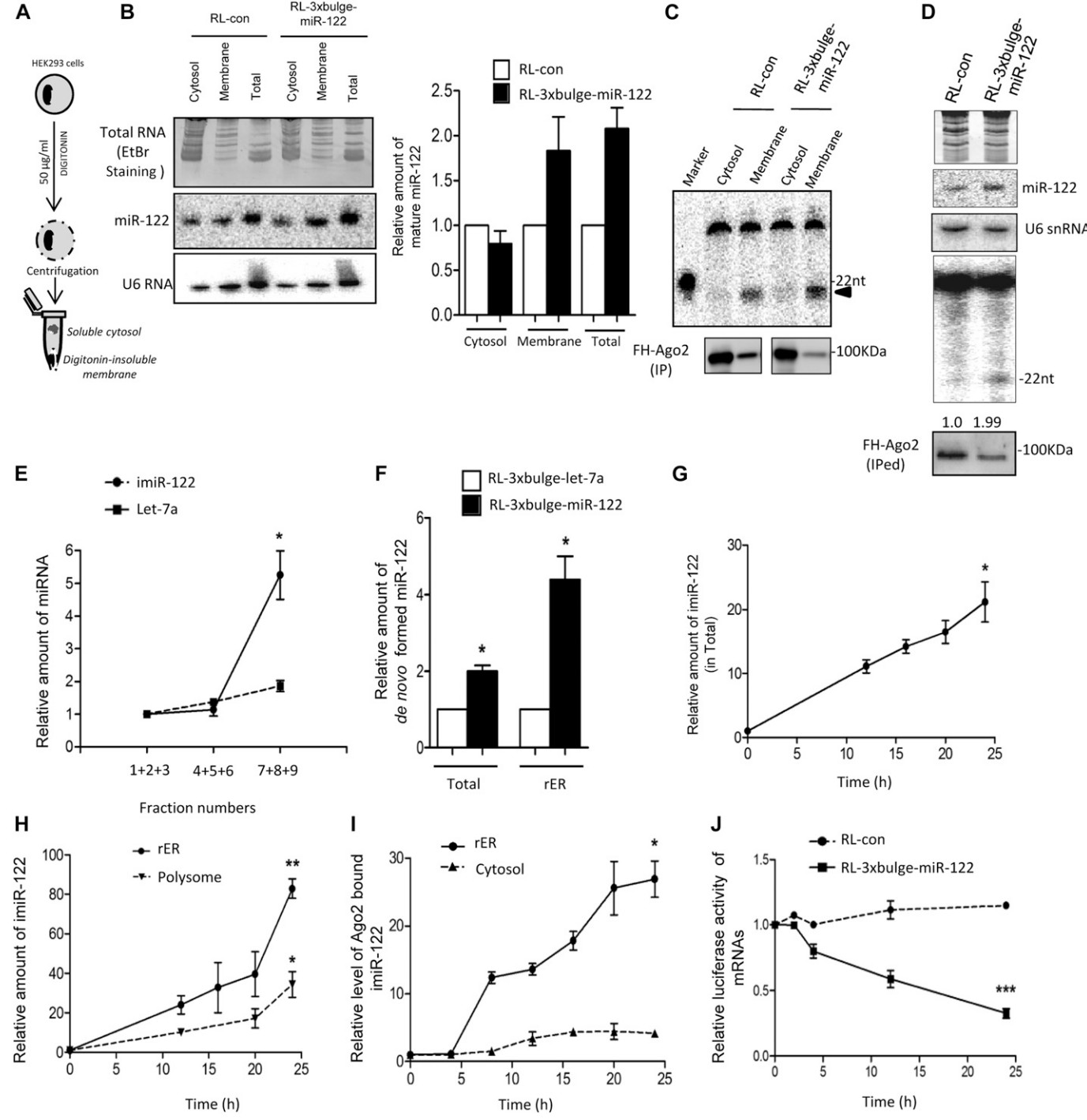

**Figure 2. Target-driven biogenesis occurs on the rER-attached polysomes.**
**(A)** Scheme of digitonin-based membrane isolation from HEK 293 cells. **(B)** Relative steady-state distribution of miR-122 in total, cytosol, and digitonin-insoluble membranes in HEK 293 cells exogenously expressing the liver miRNA-122 in the presence of its target RL-3xbulge-miR-122 and control RL-con mRNA. Northern blot (*left panel*) where ethidium bromide staining of the gel used for Northern blotting of miRNA and U6 are shown. Quantification of the same (*right panel*) against the U6 RNA has been shown. **(C)** In vitro RISC cleavage assay with FH-Ago2 IP from cytosolic and membrane fractions. Membrane-associated miRISC isolated from target RNA–expressing cells showed higher specific activity for substrate RNA cleavage. The cleaved RNA products are marked by an arrowhead. **(D)** Target mRNA–induced de novo miRNPs get enriched on microsomes. Northern blotting was used to detect mature miR-122 and U6. RISC cleavage assay was performed with protein equivalent amounts of FH-Ago2 IP. **(E)** Target mRNA–driven de novo–formed miRNAs are present in the ER fraction. Relative distribution of newly synthesized miR-122 as well as endogenous let-7a in the presence of miR-122 target mRNA. From the 3–30% OptiPrep gradient, fractions 7, 8, and 9 were pooled to get rER-associated miRNA, whereas fractions 1–3 and fractions 4–6 were enriched, respectively, for endosome/MVBs– and lysosome-associated miRNAs (as described in Fig 1C and D). Amounts of RNA used for quantification in RT-PCR reaction were identical. **(F)** Inducible miR-122 associates with microsome in the presence of target mRNA. Real-time quantification of mature miR-122 present in microsomal fraction and in total after 24 h of induction in the presence of RL-3xBulge-miR-122 and control RL-3xBulge-let-7a were measured and plotted. **(G, H)** Increase in de novo–formed mature miR-122 level with time in the presence of its target

lysate (RRL) and the reaction was reconstituted with isolated membrane or polysomes or microsomes and synthetic pre-miR-122 and target mRNA. To mimic the target-driven biogenesis under cell-free conditions using cellular membranes, we added in vitro–transcribed target mRNA and synthetic pre-miR-122 to the reaction. Membrane structure–attached polysomes were suspected to be involved in the process and were substituted in the in vitro miRNA formation reaction described above by using either digitonin-insoluble membrane or microsome or polysomes isolated from HEK 293 cells (Fig 3A). Consistent with the findings we had in vivo, we observed enrichment of target mRNAs on the membranes within 30 min of incubation in the presence of pre-miR-122 (Fig 3B). Our observation suggests that availability of precursor-miRNA leads to increased miRNP formation that may be preceded by targeting of cognate target mRNA on membrane. Target-driven biogenesis was successfully recapitulated on re-isolated membranes. With digitonin-insoluble membranes, mature miR-122 was increasingly getting accumulated with the membranes in the presence of increasing concentration of RL-3xbulge-miR-122 target mRNA (Fig 3C). Using microsome or rER, it was observed that within 30 min of the in vitro reaction, mature miR-122 and target mRNA RL-3xbulge-miR-122 get increasingly localized on the microsome against time (Fig 3D). Moreover, presence of RL-3xbulge-miR-122 was significantly higher on polysomes after in vitro translation with RRL than RL-con (Fig 3E). In the same experiment, we quantified the miR-122 level that also showed polysomal enrichment in the presence of target mRNA (Fig 3E). Consistent with the known model of target-driven miRNA biogenesis, our data revealed a time kinetics of target mRNA association and miR-122 formation on polysome. It suggests target-driven mature miRNA formation on ER-attached polysome, but the target mRNA targeting to polysomes precedes the de novo miRNA formation (Fig 3F).

## Impaired de novo miRNP formation on ER-attached polysomes in cells with depolarized and ER-detethered mitochondria

miRNA-targeted mRNAs get degraded in the LE/MVBs that are preceded by translation repression of the message on the rER (19). It has also been shown that MVB targeting of miRNPs and repressed messages is prerequisite for recycling of the exiting miRNPs engaged in repression. In the process, a large fraction of Ago2 also get decoupled from miRNAs on MVBs as the MVB-localized fraction of Ago2 is specifically miRNA deficient and phosphorylated (26). As the ER is known to be a site for RISC nucleation or Ago2 loading of siRNAs (17) and MVBs or LEs are ascertained as the region of target mRNA cleavage and Ago2-miRNA unloading (19), we were interested to find how mitochondria may have an effect on this process. It was of particular concern as it has been shown previously that mitochondria

plays an important role in controlling the subcellular compartmentalization and stability determination of miRNAs and target messages in mammalian cells (27).

In a recent report, it has also been shown how mitochondria control the interaction between the organelles to ensure miRNP recycling and how a defect in mitochondrial tethering to rER impairs the endosome–ER interaction that is associated with accumulation of miRNPs and repressed message on the rER. Protozoan parasite *Leishmania donovani* uses mitochondrial depolarization in host cells to down-regulate miRNP turnover and increases miRNP levels in infected macrophage. The increased miRNPs get accumulated with rER-associated fraction (27). In infected cells, accumulation of miRNA-free Ago proteins in the endosome/MVB fraction was also noted (27). The significance of this accumulation was not clear. Based on the observations described in earlier sections, we wanted to test whether accumulation of Ago2 protein on the endosomal fraction has anything to do with de novo miRNP biogenesis in mammalian cells having depolarized mitochondria or in a context where mitochondria is not tethered with the ER.

We used the same doxycycline-inducible expression system for expressing pre-miR-122 in HEK 293 Tet-On cells and followed the accumulation of mature and precursor-miR-122. To see the effect of mitochondrial depolarization on miRNA/pre-miR level, we have introduced uncoupler protein 2 expression plasmid, FH-Ucp2 (FLAG-HA-Ucp2), in HEK 293 Tet-On Cells (Fig 4A). In cells, transfected either with control vector plasmid or FH-Ucp2 expression plasmid in the presence of doxycycline, the time-dependent increase of miR-122 association with Ago2 was followed (Fig 4B). The de novo–mature miR-122 association with Ago2 was inhibited in cells expressing FH-Ucp2 (Fig 4B). Mfn2 is a tethering protein for mitochondria with the rER, and similar to that observed with Ucp2–expressing cells, depletion of Mfn2 by siRNA also gave a similar result of retarded miR-122 miRNP formation in cells treated with siMfn2 but not in control siRNA-treated cells (Fig 4C). Expression of FH-Ucp2 and depletion of Mfn2 were confirmed by Western blot analysis of the respective protein (Figs 4G and S2A). We could also see changes in mitochondrial morphology in those cells with punctuated, fragmented defective mitochondria compared with filamentous, extended healthy mitochondria in control cells (Fig S2B). Interestingly, subcellular fractionation of the FH-Ucp2-expressing or Mfn2-depleted cells into endosomal (early endosome) and ER fractions followed by Ago2-associated miRNA content analysis carried out for respective fractions suggested preferential depletion of de novo–formed miRNPs happened for the ER-associated Ago2 pool. Data suggests that the problem of de novo synthesis of miRNPs on the ER fraction has been affected in mitochondria-depolarized (FH-Ucp2 expressing) or ER–mitochondria–detethered cells (Fig 4D).

mRNA. Levels of miR-122 were quantified and plotted after 0, 12, 16, 20, and 24 h of induction. **(G)** Values obtained with total cellular RNA were plotted in panel (G). **(H)** Quantifications of microsome and polysome-associated miR-122 are represented in panel (H). In both cases, the level at 0 h is taken as one unit. **(I)** De novo–formed miR-122 associates with rER-bound Ago2 in the presence of target mRNA. Tet-On–stable HEK 293 cells were transiently transfected with FH-Ago2, inducible miR-122, and RL-3xbulge-miR-122–expressing plasmids. FH-Ago2 IP from cytosolic and microsome after induction were used for quantification of associated mature miR-122. In all cases, the level at 0 h is taken as one unit. **(J)** The newly formed miRNPs are functional. Repression kinetics of RL-3xbulge-miR-122 after induction of de novo synthesis of miR-122 was followed and quantified. Tet-On–stable HEK 293 cells were transfected with either RL-con or RL-3xbulge-miR-122 and inducible miR-122 expression plasmids. After 0, 2, 4, 12, and 24 h of induction for miR-122 expression, Renilla Luciferase expression in cells transfected with RL-con or RL-3xbulge-miR-122 was measured in the dual luciferase assay. Firefly luciferase–expressing plasmids were co-transfected in all cases, and relative luciferase activity in 0 h was taken as unit. In all cases, Renilla luciferase (RL) expression were normalized by firefly expression. Paired two-tailed *t* tests were used for all comparisons. *P* < 0.05 (*); *P* < 0.01 (**); *P* < 0.001 (***). For statistical analysis, minimum three sets of data were used. Source data are available for this figure.

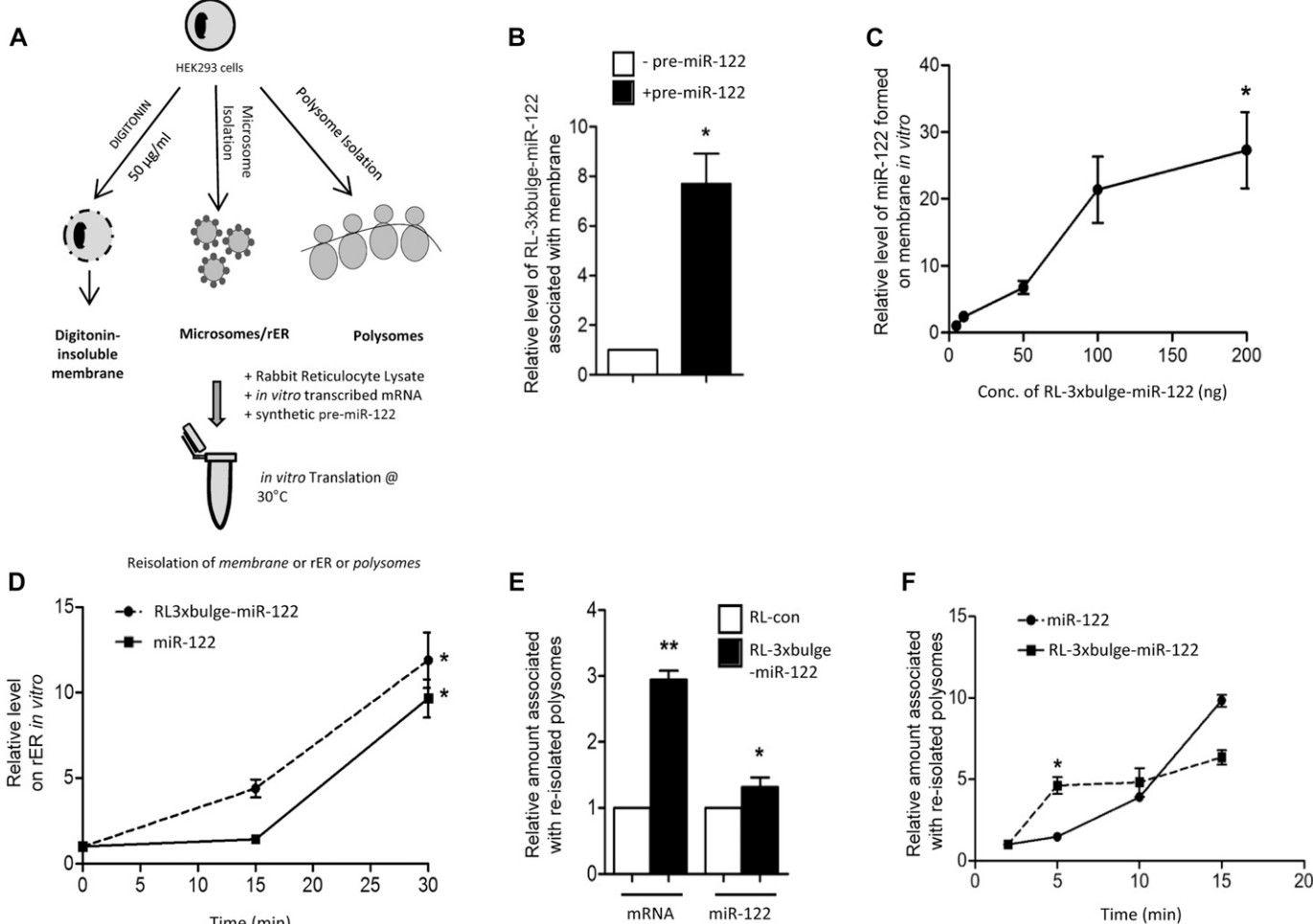

**Figure 3. Target mRNA–dependent miRNA formation on isolated cellular membranes in vitro.**
**(A)** Target-dependent miRNA biogenesis assay in vitro. Schematic representation of the experiment is shown. Digitonin-insoluble membrane was incubated with rabbit reticulocyte lysate (RRL), synthetic pre-miR-122, and in vitro–transcribed mRNAs (50 ng) and subjected to in vitro translation reaction for 30 min at 30°C. After the reaction, the membranes were reisolated and mRNA levels associated with the membrane were quantified. **(B)** Enrichment of target mRNA in the presence of cognate miRNA on digitonin-insoluble membrane in vitro. **(C)** Effect of target mRNA concentration on miRNA biogenesis in vitro. The translation reaction with RRL was performed with isolated digitonin-insoluble membranes and with increasing concentration of in vitro transcribed RL-3xbulge-miR-122 (0, 10, 50, 100, and 200 ng) mRNA and synthetic pre-miR-122. After the reaction, membrane-associated mature miR-122 formed was quantified. **(D)** Increased association of RL-3xbulge-miR-122 and mature miR-122 with microsome in vitro. In vitro translation was performed with isolated microsome and RRL, pre-miR-122, target RNA for different time points (0, 15, and 30 min) and levels of mature miR-122 formed and attached with microsome were quantified after reisolation of microsome from the reaction. **(E)** Polysomal enrichment of target mRNA and its cognate miRNA in the in vitro translation cum miRNA biogenesis assay system. Polysome association of mRNA and miR-122 after in vitro translation for 30 min carried out with polysome isolated from HEK 293 cells in the presence of synthetic pre-miR-122 and in vitro–transcribed RL-con or RL-3xbulge-miR-122. **(F)** Targeting of mRNA to polysome precedes cognate miRNA biogenesis and its association with polysome in vitro. Time-dependent polysome association of target mRNA and de novo–synthesized cognate miRNAs with time. Translation reaction was carried out as described in previous panels, in the presence of RL-3xbulge-miR-122 and pre-miR-122, in a reaction mixture containing polysomes and RRL for different time intervals, and amounts of miRNA and mRNA associated with polysomes reisolated after the reaction were quantified and plotted. All experiments were carried out three times. Paired two-tailed t tests were used for all comparisons. $P < 0.05$ (*); $P < 0.01$ (**); $P < 0.001$ (***).

It has been reported previously that the re-feeding of amino acid–starved hepatic cells is associated with de novo synthesis of miR-122 (15) (Fig 4E). In this physiological context, we also explored the effect of mitochondria depolarization on de novo miR-122 production in re-fed cells upon amino acid starvation. Consistent with our observation in HEK 293 cells, where ectopic expression of mature miR-122 miRNPs was prevented in presence of excess Ucp2, we have noticed a similar decrease in mature miR-122 production in re-fed Huh7 cells expressing FH-Ucp2 (Fig 4F). It was consistent with an increase in miR-122 target mRNA levels in FH-Ucp2 expressing cells (Fig 4F). It was also consistent with the

increase in pre-miR-122 level and drop in the amount of miRNA produced per molecule of target mRNAs in re-fed cells expressing FH-Ucp2 (Fig 4H).

### ER targeting of mRNA remains unaffected in cells with mitochondria-detethered ER

It has been reported previously that targeting of mRNAs to ER-attached polysomes is prerequisite for their interaction with the Ago protein (18). It was also known that the miRNA-loaded Ago protein remains bound to Dicer1 before it interacts with the target

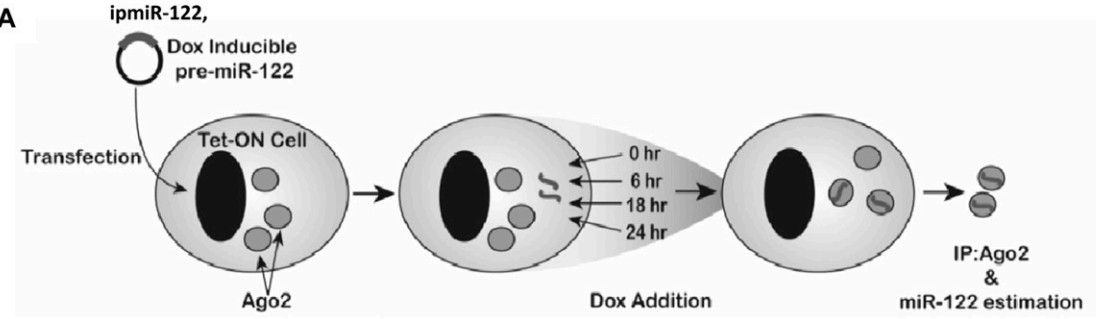

**Figure 4. Defective de novo miRNP formation in mitochondria-depolarized cells.**
**(A)** Scheme of the experiment used to determine the de novo rate of Ago2–miR-122 miRNP formation using doxycycline-inducible Tet-On HEK 293 cells. The Tet-ON HEK 293 cells were transfected with doxycycline-inducible pre-miR-122 plasmid construct (ipmiR-122). Doxycycline was added to the cells for 48 h after transfection. The cells were harvested after doxycycline induction of indicated time periods, and immunoprecipitation was performed for endogenous Ago2 protein. miRNA estimation and relative quantification were performed for IPed Ago2-associated miRNA by qRT-PCR. **(B, C)** Values of Ago2-associated miR-122, mature miR-122, or precursor-miR-122 in Tet-on HEK 293 cells induced with doxycycline for specified time periods. Shown are the mean and SEM values from at least three independent experiments. **(B, C)** All cells were co-transfected with Renilla-3xbulge-miR-122 reporter, Tet-inducible pre-miR-122 expression plasmid ipmiR-122, and FH-Ucp2 (panel B) or siMfn2 (panel C). IPed Ago2 level, U6 small nuclear RNA level, and 18S ribosomal RNA level were used as internal controls for the estimation of Ago2-associated miR-122, mature miR-122, and precursor-miR-122, respectively. All values have been normalized against values at 0 h. **(D)** Shown are the mean and SEM from at least four independent experiments with values of Ago2-associated miR-122 in an iodixanol OptiPrepR (3–30%) gradient fractions positive for early endosome (EE) and ER markers. Lysate was obtained from TET-ON HEK 293 cells induced with doxycycline for 24 h. All cells were co-transfected with Renilla-3xbulge-miR-122 reporter, Tet-inducible pre-miR-122 expressing ipmiR-122, and either FH-Ucp2 or siMfn2. miR-122 present in ER fractions from each set were normalized against corresponding EE-associated miR-122 levels. **(E, F)** Target mRNA–driven miRNA biogenesis is hampered in mitochondria–ER uncoupled condition in Huh7 cells. **(E)** Scheme of the experiment is shown in panel (E). qRT-PCR data have confirmed no up-regulation of miR-122 biogenesis in the presence of its target under Ucp2 over-expression condition during amino acid re-feeding (F, left panel). *CAT-1* mRNA level was also increased after 4 h of amino acid starvation on HA-Ucp2–expressing cells (F, right panel). **(G)** Western Blot data confirm HA-Ucp2 over-expression on Huh7 cells. **(H)** Pre-miR-122 expression in starved versus re-fed conditions in HA-Ucp2–expressed cells. Reduced pre-miRNA processing was observed upon

mRNAs 3'UTR. Subsequently, the Ago2-miRNA-Dicer1 complex scans the 3'UTR to find the target site(s) and release the Dicer1 for a new round of pre-miRNA processing (15). Therefore, interaction of the mRNA with the miRNPs on the rER could be a rate-limiting step in the target-driven cognate miRNA biogenesis happening on the rER of mammalian cells.

Interestingly, we documented an impairment of de novo miRNP biogenesis in MEF cells derived from Mfn2$^{-/-}$ mouse embryo compared with MEF taken from a wild-type animal embryo expressing Mfn2 (Fig 5A). This could be due to impaired targeting of the target messages to the ER membrane causing a defect in miRNA biogenesis. Increased steady-state expression of miR-122 and repression of its target mRNA has been documented in the MEF cells without Mfn2 (Fig 5B and C). This is consistent with increased steady-state levels of miRNPs reported in Mfn2$^{-/-}$ cells (27). Interestingly, increased targeting of target mRNAs to the rER or microsome in Mfn2$^{-/-}$ MEF cells has also been noted (Fig 5D and E). These data suggest increased ER targeting of mRNA in Mfn2$^{-/-}$ cells, contrary to what suspected to be the rate-limiting step for lower target-driven miRNA biogenesis in Mfn2$^{-/-}$ cells. To test the point further, we used an in vitro assay where isolated microsome was incubated, either with crude mitochondrial fractions derived either from wild-type or Mfn2$^{-/-}$ cells, in the presence of target mRNA and pre-miR-122 (Fig 5F). After the reaction, association of target mRNAs and de novo–synthesized miRNA-122 with the microsome was measured. Consistent with in vivo data, we observed an increased association of miR-122 target mRNA with the microsome incubated with Mfn2$^{-/-}$ mitochondria (Fig 5G). However, there was a reduction in de novo miRNA formation and its association with microsome in unit time in presence of the target mRNAs and incubated with Mfn2$^{-/-}$ mitochondria (Fig 5H). This result rules out the possibility of impaired target message association to the rER and polysomes as the cause of retarded de novo miRNP formation observed in Mfn2$^{-/-}$ cells.

### Defective compartmentalization of Ago2 and Dicer1 causes restricted de novo miRNP formation in Mfn2$^{-/-}$ cells

Fractionation of the MEF cells into different subcellular compartments and subsequent analysis of the contents of different fractions revealed predominant association of Ago2 and Dicer1 with the endosomal fraction in Mfn2$^{-/-}$ MEF cells (Fig S3A–C). From a microscopic study carried out in the Mfn2$^{-/-}$ MEFs, we also observed increased signals both for Ago2 and Dicer1 with the endosomal compartments (Figs 6A–C and S3D–F). These data were consistent with previous observations made with cells expressing the mitochondrial membrane potential decoupling protein Ucp2 in excess or in cells where the pathogen *L. donovani* induces depolarization of mitochondria (28). In that work, retarded organelle dynamics and mitochondrial detethering with ER and endosomes were also noted, resulting in lowered juxta-positioning of the ER and endosomes in mitochondria-compromised cells. Interestingly, lower juxta-positioning of the ER and endosomes was also substantiated in Mfn2$^{-/-}$ MEFs (29). Therefore, it is possible that this loss of

endosome–ER contacts resulted in higher accumulation of Ago2 and Dicer1 on endosomes that are needed to be trafficked back to the ER for its loading with de novo miRNAs and formation of new miRNPs in a target RNA–dependent manner on the ER-attached polysome. To support the notion, we additionally performed in vitro Ago2 trafficking experiments with endosomes purified from the HEK 293 cells expressing the FH-Ago2. Endosomes were incubated in the in vitro reaction in the presence of rER fraction isolated either from the wild-type or Mfn2$^{-/-}$ MEF cells. Usually, the rER is tethered to the mitochondria, and the process is compromised in Mfn2$^{-/-}$ cells. After the incubation, the ER fraction was re-isolated, and its association with transferred Ago2 was estimated. Interestingly, the transfer of Ago2 to the rER from the endosomal fraction was higher in the wild type than in Mfn2-negative samples (Fig 6D–F). Deficiency of miRNA biogenesis observed in the Mfn2$^{-/-}$ cells can, thus, be corroborated with the in vivo observations we had earlier. The importance of miRNA-unloaded Ago2 trafficking to the ER for de novo miRNP formation was further supported when partial restoration of miRNP biogenesis could be achieved in Mfn2-deficient cells expressing HA-Ago2 in excess but not in cells expressing HA-Ago2Y529E, an Ago2 mutant with mutation in its MID domain that restricts its miRNA binding (Fig 6G) (30).

In the previous work, it has been described that most of the Ago2 present in the endosomal fraction is not loaded with miRNAs (26), and it has also been reported previously that Ago2 miRNA loading is dependent on its interaction with Dicer1 and also on the phosphorylation status at the Y529 position as the phosphorylation of Ago2 at this site leads to its unbinding of miRNAs (31). Upon immunoprecipitation performed with total cell extract using Ago2-specific antibody, we observed a retarded interaction of Dicer1 and Ago2 protein in Mfn2-negative MEFs, and the data were consistent also with increased level of phosphorylation of Ago2 in the endosome fraction of Mfn2$^{-/-}$ cells (Fig 6H–J). Therefore, the detethering of mitochondria with other organelle induces an abortive Ago2–Dicer1 complex formation in Mfn2-depleted cells along with increased Ago2 phosphorylation that may contribute further down-regulation of de novo miRNP formation in Mfn2-negative cells.

## Discussion

In this article, we have shown the importance of organelles in controlling the miRNP biogenesis process. Consistent with the previous predictions (19, 27), we have strong evidences in support of ER-attached polysomes as the primary site for miRNP biogenesis in mammalian cells. It has also become evident that miRNP biogenesis is a regulated process and is sensitive to cellular energy status that is primarily governed by mitochondrial activity (Fig 7). By ensuring the inter-organelle interactions between two separate cellular domains, mitochondria may ensure exchange of their components such as the Ago protein and Dicer1 to ensure regulated

re-fed of Ucp2 over-expressed cells compared with control. Paired two-tailed *t* tests were used for all comparisons. *P* < 0.05 (*); *P* < 0.01 (**); *P* < 0.001 (***). In (A, B, C, D, E, F, G, and H), values are means from at least three biological replicates ± SEM.
Source data are available for this figure.

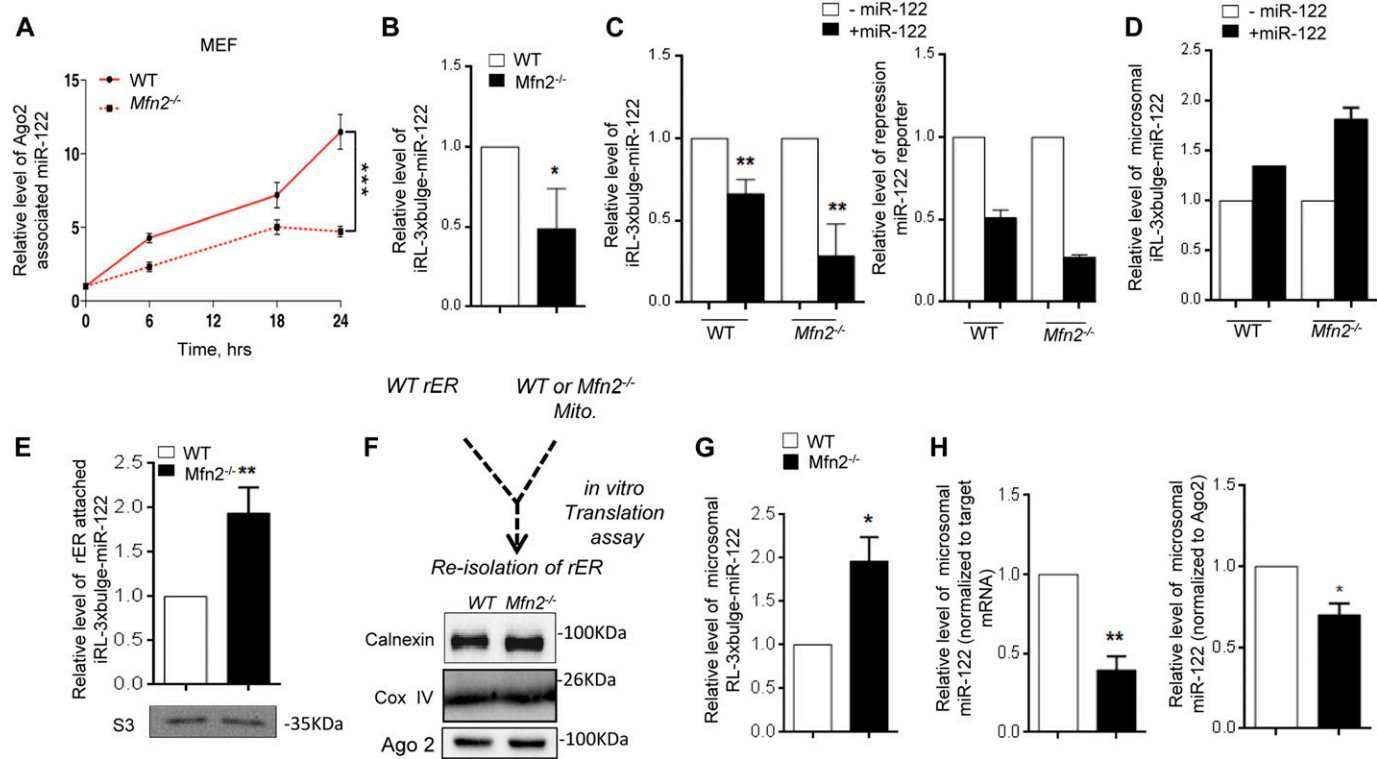

**Figure 5. Increased target RNA trafficking to rER in Mfn2 negative cells.**
**(A)** Decreased association of newly formed miR-122 with Ago2 on mitochondria-defective cells. qRT-PCR data showing Ago2-associated de novo formed miR-122 level decrease over time on Mfn2$^{-/-}$ cells compared with WT cells was plotted. **(B)** Decreased de novo–formed target mRNA in Mfn2$^{-/-}$ MEF cells. qRT-PCR data show reduced target mRNA levels in the presence of its cognate miRNA on Mfn2$^{-/-}$ cells. **(C)** The relative amount of the reporter target mRNAs in the absence and presence of miR-122 has been shown in both cell types. All cells were co-transfected with Tet-ON expression plasmid and inducible Renilla-3xbulge-miR-122 reporter plasmid and FF reporter plasmid. In the right panel, relative fold repression of the miR-122 reporter has been studied in the presence and absence of miR-122 expression in wild-type and mitochondria-defective cells. **(D, E)** Higher microsomal association of target mRNA in Mfn2$^{-/-}$ MEF cells. Microsomal sequestration of target mRNA in the presence and absence of its cognate miRNA has been plotted (D). The relative level of microsome/rER-associated target mRNA between normal and Mfn2-negative cells expressing pmiR-122 has also been shown (E). Western blot data of ribosomal protein S3 show equal amounts of microsome associated ribosomes were used for estimation of associated mRNAs. **(F)** Schematic representation of in vitro translation assay showing reisolation of rER used for the assay performed in the presence and absence of wild-type and mutant cell derived mitochondria. Calnexin, Ago2, and COX IV Western blots confirmed equivalent amount of microsome along with mitochondria after reisolation. **(G)** Relative estimation of RNA from reisolated microsome after incubation in the presence of mitochondria obtained from wild-type and Mfn2$^{-/-}$ cells. Increased target mRNA sequestration in the presence of cognate miRNA in assays performed with detethered mitochondria was observed. **(H)** Measurement of miR-122 being formed and became associated with microsome showed reduced miR-122 association for per unit of RL-3xbulge-miR-122 mRNA when rER from wild-type cells were incubated with Mfn2$^{-/-}$ mitochondria (left panel). Reduced microsomal association of miR-122 has been also observed when normalization was done with endogenous control Ago2 (right panel). In both panles while bars represent WT and black bars denote Mfn2$^{-/-}$. Paired two-tailed *t* tests were used for all comparisons. $P <$ 0.05 (*); $P < 0.01$ (**); $P < 0.001$ (***). In (A, B, C, D, E, and F), values are means from at least three biological replicates ± SEM. Source data are available for this figure.

formation of de novo miRNPs to enhance effective gene repression process.

miRNPs are believed to be very stable molecules that also undergo turnover associated with extracellular vesicles-mediated export of Ago2-uncoupled miRNAs from the LEs and MVB compartments. However, the fate of the "miRNA-free" Ago2 is not clear. Although a fraction of the Ago2 may, as a part of quality control check, undergo degradation upon targeting of Ago2 to lysosomal compartment (19, 32), a fraction of the same is available and get trafficked back to ER for its loading with new mature miRNAs processed by associated Dicer1 on the polysome attached with the ER.

It is an important question why the miRNA-mediated processes are compartmentalized in mammalian cells. Although the clear answer is still unavailable, it can be explained by compartment-specific miRNA activity regulation. While the translation

repression machineries are mostly associated with rER, the miRNA-mediated target degradation and associated miRNP turnover are heavily regulated at MVB/P-body compartments. Thus, these two spatiotemporally uncoupled processes become independent in mammalian cells (19) to ensure the required robustness of miRNA-mediated gene repression processes in mammalian cells. The two pools of Ago proteins are also different from each other. Whereas the ER associated Ago proteins are primarily loaded with miRNAs and as expected are in bound forms with their targets (26), the MVB located Ago proteins are phosphorylated and primarily unbound to miRNAs and target mRNAs. The MVB-localized Ago can also interact with Dicer1 and, therefore, it is possible that the Dicer1–Ago interaction may happen irrespective of their binding with pre-miRs and mature miRNAs. It is to be noted that both mature and pre-miRNAs are accumulated on the rER, and therefore, the processing of pre-miRNAs and subsequent transfer to Ago2 may occur by

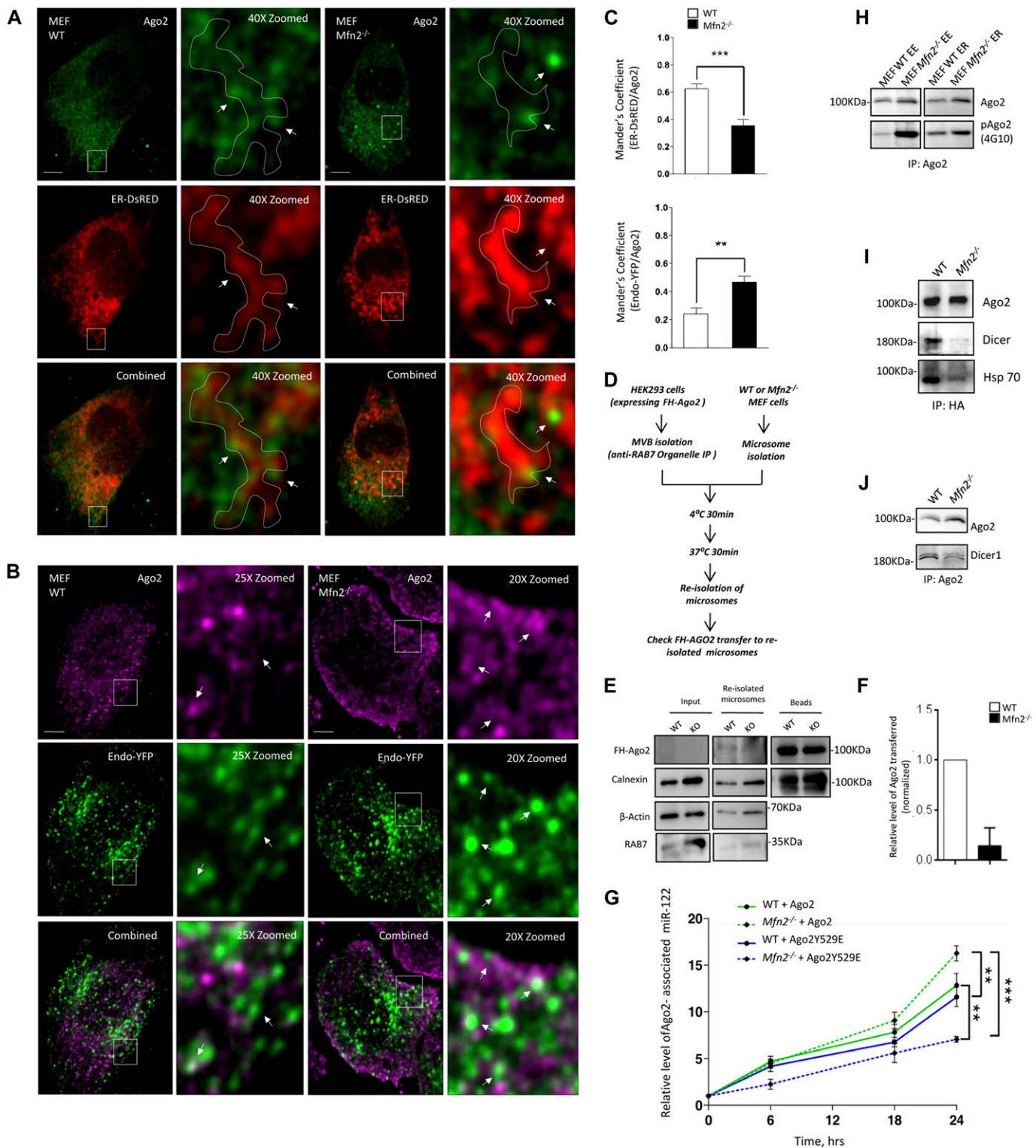

**Figure 6. Retarded Ago2 retro transport to rER-attached polysome in cells with defective ER–endosome interaction.**
**(A, B, C)** Representative combined frames showing endogenous Ago2 protein localization with ER (A) or endosomes (B) in MEFs of depicted genotypes. In panel (A), cells expressing an ER-targeting variant of DsRed (ER-DsRed, *red*) were stained by indirect immunofluorescence for endogenous Ago2 (*green*). Likewise, in *panel (B)*, cells expressing an endosomal targeting variant of YFP (Endo-YFP, green) were stained by indirect immunofluorescence for endogenous Ago2 (*purple*). Endo-GFP or ER-dsRED signals were detected by direct fluorescence of the tagged protein. Marked areas are zoomed and ER region of interest in the zoomed images has been enclosed by a dotted perimeter. White arrows depict foci with either elevated or reduced colocalization between ER or Endosomal structures with Ago2 bodies. Scale bars, 10 $\mu$m. Also shown are the mean ± SEM from at least four independent experiments (10 cells per experiment) of colocalization data between ER-Ago2 (C, *top*) and Endosome-Ago2

Ago-associated Dicer1 even after the miRNA-independent interaction of Ago–Dicer1 complex with the 3′UTR of target mRNAs (15). Therefore, compartmentalization of Ago proteins is a very effective way to control miRNA biogenesis and recycling.

From the observations we made, it seems that the process of Ago2 trafficking from endosomal compartment to ER compartment is blocked in mitochondria-defective cells that also get recapitulated in an cell-free in vitro Ago2 transfer assay adopted in this study. Therefore, the energy status and other mitochondrial activity changes can influence the miRNP biogenesis primarily by affecting the intracellular Ago2 shuttling. Involvement of mitochondria in controlling the miRNA turnover has been documented previously (27). Here, we have gathered evidence to claim that the biogenesis process of mature miRNAs is heavily dependent on mitochondria. It has been shown earlier that the metabolic flux is controlled by mitochondria, and that the same is also controlled by miRNAs (33, 34). This holds true also for autophagy regulation both by mitochondria and miRNAs (35, 36). Therefore, some of the observed effects associated with mitochondrial abnormalities may be attributed indirectly by associated defective miRNA biogenesis in different model systems of mitochondrial dysfunction. The mechanistic detail of how mitochondria affects the Ago protein–Dicer1 complex formation is, however, not clear. It is possible that some "hot-spots" on the MVB membrane are the sites for Ago–Dicer1 interaction, and these missing in mitochondria defective cells, are prerequisite for miRNP formation happening on ER-attached polysomes. Mitochondrial tethering with the ER membrane may have a major role to play in controlling the miRNP formation by indirectly affecting the Ago2 compartmentalization process, and miRNA biogenesis is sensitive to mitochondrial detethering with the ER. However, the exact mechanism is yet to be unraveled.

# Materials and Methods

### Plasmids and siRNAs

The miR-122 expression from an encoding plasmid has been performed as described elsewhere (37). The Renilla reporter plasmids containing humanized Renilla Luciferase CDS (RL-con) or with three miRNA-binding sites (either for miR-122 or let-7a) downstream of the Renilla Luciferase (RL)–coding region, RL-3xbulge-miR-122 or RL-3xbulge-let-7a (kind gifts from Witold Filipowicz), were used. pTet-On Advanced Vector (Clontech) was used to make TET-inducible stable HEK 293 cells. The inducible expression constructs, iRL-3xbulge-miR-122 and imiR-122, have also been described earlier (15, 18).

### Cell culture and transfection

Human HEK 293 and MEF cells were grown in DMEM containing 2 mM L-glutamine and 10% heat-inactivated FCS. All plasmid transfections were performed using Lipofectamine 2000 and siRNAs (at 50 nM concentration) using RNAiMax (Life Technologies) as per the manufacturer's protocol. For experiments using tetracycline-inducible constructs, induction was performed using 300 ng/ml of doxycycline (Sigma-Aldrich).

### Luciferase assay

Luciferase assays were performed using Dual-Luciferase Assay Kit (Promega), following the manufacturer's instructions, on VICTOR X3 Plate Reader with injectors (PerkinElmer). The ratio of firefly luciferase–normalized RL expression levels in control to experimental sets were measured as fold repression.

### RNA isolation, Northern blotting, and real-time PCR analysis

TRIzol reagent (Life Technologies) was used to isolate total RNA. Northern blotting was performed as described early (38). For detection, $\gamma^{32}$P-labeled 22-nt miRCURY complementary LNA probes for miR-122/let-7a (Exiqon) or cDNA probe for U6 small nuclear RNA were used. Blots were detected using phosphor imaging by Cyclone Plus Storage Phosphor System (PerkinElmer), and ImageJ was used for quantification. In case of qRT-PCR, miRNA quantification was performed by TaqMan-based miRNA assays (Applied Biosystems) following the manufacturer's instructions. mRNA qRT-PCR was carried out using Eurogentec Reverse Transcriptase Core Kit as per the manufacturer's instructions. cDNA was prepared using random nonamers and PCR with gene-specific primers using MESA GREEN pPCRMaster Mix Plus (Eurogentec).

---

(C, *bottom*). Mander's coefficient was used to represent the colocalization between organelles under consideration and Ago2 foci. **(D, E, F)** Scheme of the in vitro inter-organellar Ago2 transfer assay is shown in panel (D). Microsome isolated form WT or Mfn2$^{-/-}$ MEF cells (designated as WT and KO) were incubated with MVBs isolated form FH-Ago2–expressing HEK 293 cells and subsequently the microsome was re-isolated. MVB isolation was performed by immunoprecipitation using anti-Rab7 antibody, and the subsequent incubation with microsome was performed on the beads bound to MVBs. Western blot data indicated transfer of FH-Ago2 from MVBs (HEK 293) to the microsome isolated from WT MEF cells, whereas Mfn2 knockout prevents transfer of FH-Ago2. Quantification of the same is shown in panel (F). Marker protein calnexin was used to indicate equality of re-isolated microsome content (E). **(G)** Level of Ago2-associated doxycycline-induced miR-122 at specified time points in MEFs of indicated genotype. All cells have been co-transfected with Tet-ON–expressing Renilla-3xBulge-miR-122 reporter and Tet-inducible pre-miR-122–expressing ipmiR-122 and complemented with either HA-Ago2 or HA-Ago2Y529E expression plasmids as indicated. miR-122 levels were normalized against corresponding values at 0 h. Shown are the mean and SEM from at least three independent experiments. *P*-values were calculated by *t* test, and one, two, and three asterisks represent *P*-values less than 0.05, 0.01, and 0.001, respectively. **(H)** Phosphorylation level of Ago2 was determined in subcellular fractions from MEFs of indicated genotypes. OptiPrep (3–30%) density gradients were used for separation of fractions positive for EE and ER and were used for endogenous Ago2 protein immunoprecipitation. The level of phosphorylated-Ago2 (4G10) present in IP Ago2 from ER and EE positive fractions was determined by Western blotting. **(I, J)** Immunoprecipitation of FH-Ago2 was performed with anti-HA antibody to pull down Ago2 from MEFs of indicated genotype transfected with Flag-HA tagged Ago2 (FH-Ago2). Western blot analysis was performed for indicated proteins to check their association with IPed Ago2 (*panel I*). In panel (J), antibodies against endogenous Ago2 was used to immunoprecipitate Ago2 to confirm its Dicer1 dissociation in MEF cells. Paired two-tailed *t* tests were used for all comparisons. *P* < 0.05 (*); *P* < 0.01 (**); *P* < 0.001 (***). In (A, B, C, D, E, F, G, H), values are means from at least three biological replicates ± SEM.

Source data are available for this figure.

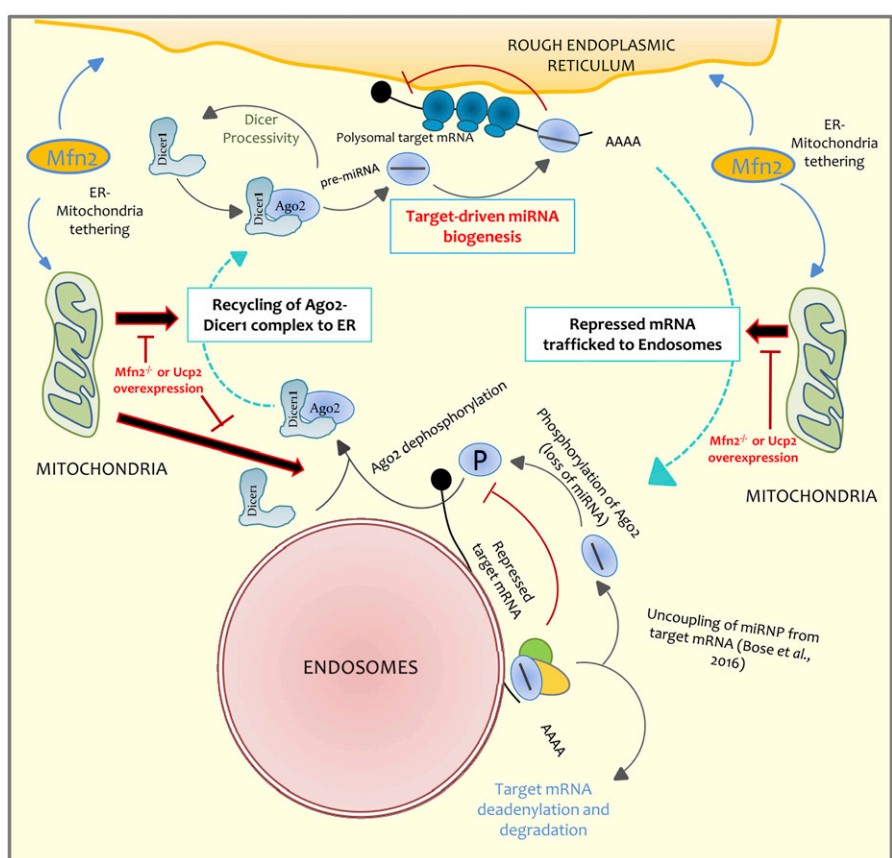

**Figure 7. Ago2 recycling determined by mitochondria controls Ago2–miRNA complex formation on the polysome attached to the rER.**
In this schematic model, the mode of miRNA biogenesis regulation by mitochondria is shown. Polysome attached to the ER serves as the cellular sites where this event occurs. This is an unique example of organellar control of a posttranscriptional gene regulatory process where the components of miRNP complexes are differentially localized to the ER and endosome-associated compartments, and internal exchange of Ago2 between compartments by mitochondria-driven inter-organellar interactions controls the rate-limiting step of miRNP formation process.

## Immunoprecipitation and Western blotting

For immunoprecipitation of FLAG-tagged Ago2 from HEK 293 cells, FLAG-M2 Agarose beads (Sigma-Aldrich) were used as described previously (25). Proteins and RNAs were extracted from beads for further analysis. For Western blotting, proteins were transferred to polyvinylidene fluoride or polyvinylidene difluoride membrane after SDS–PAGE electrophoresis followed by blocking and probing with either of the following primary antibodies (dilution mentioned also) at 4°C overnight. Mouse anti-Ago2 (Abnova) (1:1,000), Rabbit anti-calnexin (Cell Signaling Technology) (1:1,000), Rabbit anti-HRS (Bethyl Laboratories) (1:1,000), Rat anti-HA (Roche) (1:1,000), HRP-conjugated $\beta$–Actin (Sigma-Aldrich) (1:10,000), HRP-conjugated GAPDH (Sigma-Aldrich) (1:50,000), Rabbit anti-COX4 (Cell Signaling Technology) (1:1,000), Rabbit-LAMP 1 (Cell Signaling Technology) (1:1,000), Mouse-Alix (SantaCruz) (1:200), Mouse-P-Ago2 (4G10) (Millipore) (1:1,000), and Rabbit anti-S3 (1:1,000) were used with dilutions mentioned here.

## Cell fractionations

Cells were treated with 50 $\mu$g/ml digitonin (Calbiochem) for 10 min at 4°C followed by centrifugation at 2,500$g$ as described (18). 3–30% OptiPrep (iodixanol) density gradient centrifugation was carried out as described elsewhere (39). Polysome and microsome as well as KCl–puromycin extraction of microsome was carried out following

the protocol described earlier (18). After isolation of respective fractions, equivalent amount of RNA was used to check organellar compartmentalization of miRNAs or target mRNAs over total cellular fractions. To check differential compartmentalization of proteins, equivalent amount of proteins from organellar fractions were run for Western blots compared with total cell lysate.

## Crude mitochondrial fraction isolation

Cells were resuspended in 1× hypotonic buffer (10 mM Hepes, pH 7.8, 1 mM EGTA, and 25 mM KCl) equivalent to three times the packed cell volume and incubated for 20 min on ice. The cells were spun down and resuspended in 1× isotonic buffer (10 mM Hepes, pH 7.8, 1 mM EGTA, 25 mM KCl, and 250 mM Sucrose) twice the packed cell volume and homogenized. The cell lysate was pre-cleared at 1,000$g$ for 10 min twice to remove nuclear fractions. Then, the supernatant was spun at 12,000$g$ for 10 min to get "Crude Mitochondrial Fraction."

## In vitro translation assay

Microsome and crude mitochondrial fraction were isolated as described above and protein-equivalent microsomal and mitochondrial fractions were incubated in the presence of RRL in an in vitro translation system along with in vitro–synthesized mRNA (RL-control or RL-3xbulge-miR-122) of 200 ng or 200 pmol of Pre-miR-122 in 10 $\mu$l reaction for 30 min at 30°C.

### In vitro RISC cleavage assay and data analysis

In vitro RISC assay was performed exactly as described earlier (39). FH-Ago2 was immunoprecipitated from the individual fractions, eluted with 3× FLAG peptide into 100 $\mu l$ solution. An aliquot of 20 $\mu l$ was Western blotted for FH-Ago2 levels. The bands were quantified by ImageJ, and volume equivalent amounts of the eluted FH-Ago2 from each fraction were used for RISC cleavage assay.

Fractions of the cleaved product after in vitro RISC cleavage reaction performed with FH-Ago2 present in each subcellular fraction were measured. To calculate the specific activity of FH-Ago2 present in that fraction, total RISC cleavage activity was divided by relative amount of Ago2 present in each reactions. For drawing the graph, sum of RISC activity present over all fractions was taken as 1.

To calculate relative specific activity of miRISCs, volume equivalent amounts of FH-Ago2 immunoprecipitated from the individual fractions were subjected to RISC cleavage assay as well as Western blotting analysis. The percent cleaved product from the autoradiogram was normalized to the immunoprecipitated FH-Ago2 amounts and relative specific activities of FH-Ago2 in each fraction were measured.

### Organelle immunoprecipitation and in vitro inter-organelle Ago2 transfer assay

In vitro RISC assay was performed exactly as described earlier (39). Organelle immunoprecipitation was performed as described previously (27). Anti-Rab7 antibody was used to immunopurify LEs/MVBs from 3 to 30% iodixanol gradient fractions rich for these organelles of HEK 293 cells stably expressing FH-Ago2. For the in vitro inter-organelle Ago2 transfer assay, approximately $2 \times 10^7$ FH-Ago2–expressing HEK293 cells were used for organelle immunoprecipitation of MVBs, using anti-Rab7 antibody. Microsome was also isolated from WT or $Mfn2^{-/-}$ MEF cells. The isolated MVB on anti-Rab7–bound protein G Sepharose beads were incubated with microsome from WT or $Mfn2^{-/-}$ MEF cells (pre-cleared with protein G Sepharose beads), for an initial 30 min at 4°C followed by another 30 min at 37°C. Thereafter, microsome was re-isolated from the supernatant by $CaCl_2$ precipitation. Western blotting was performed to check if FLAG-tagged Ago2 was being transferred from MVBs to microsome.

### Immunofluorescence and image analysis

For immunofluorescence, the cells were transfected with 250 ng of Ago2 or Dicer1 expression plasmids in a six-well format. The cells were split after 24 h of transfection and subsequently subjected to specific experimental conditions. For immunofluorescence analysis, the cells were fixed with 4% paraformaldehyde for 30 min, permeabilized, and blocked with PBS containing 1% BSA and 0.1% Triton X-100 and 10% goat serum (GIBCO) for 30 min. After incubation with primary antibodies in the same buffer of desired dilution overnight at 4°C and subsequent washing steps, secondary anti-rabbit or anti-mouse antibodies labeled either with Alexa Fluor 488 dye (green), Alexa Fluor 594 dye (red), or Alexa Fluor 647 dye (far red) fluorochromes (Molecular Probes) were used at 1:500

dilutions. After 2 h of incubation at 37°C for 1 h followed by washing steps, the cells were mounted with Vectashield with DAPI (Vector Lab, Inc) and observed under Plan Apo VC 60×/1.40 oil or Plan Fluor 10×/0.30 objectives on an inverted Eclipse Ti Nikon microscope equipped with a Nikon Qi1MC or QImaging-Rolera EMC$^2$ camera for image capture. Few images were taken also on Zeiss Confocal Imager LSM800.

For image analysis, all Western blot images were processed with Adobe Photoshop CS4 for all linear adjustments and cropping. All images captured on Nikon Eclipse Ti microscope or LSM800 microscope were processed with Nikon NIS ELEMENT AR 3.1 software or with IMARISx64 software developed by BITPLANE AG Scientific software. Image cropping was performed using Adobe Photoshop CS4. All graphs and statistical analyses were generated in GraphPad Prism 5.00 (GraphPad). Two sample $t$ test was used for analysis. $P$-values < 0.05 were considered to be statistically significant and >0.05 were ns. Error bars indicate mean ± SEM.

## Supplementary Information

## Acknowledgements

We thank Witold Filipowicz and Gunter Meister for different constructs used in this study. We thank the Funding body, Department of Science and Technology (DST), Government of India, and Council of Scientific and Industrial Research. M Bose, Y Chakrabarty, B Barman, and S Chatterjee received their fellowship from Council of Scientific and Industrial Research, whereas SN Bhattacharyya was supported by SwarnaJayanti Fellowship (DST/SJF/LSA-03/2014-15) and a High Risk High Reward Grant (HRR/2016/000093) from DST.

### Author Contributions

M Bose: data curation, validation, and investigation.
S Chatterjee: validation and investigation.
Y Chakrabarty: resources, validation, and visualization.
B Barman: validation, investigation, and methodology.
SN Bhattacharyya: conceptualization, data curation, and investigation.

### Conflict of Interest Statement

The authors declare that they have no conflict of interest.

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
