## [Reviewer comments · Life Science Alliance]

Life Science Alliance

Retrograde Trafficking of Argonaute 2 Acts as a Rate-Limiting Step for de novo miRNP Formation

Suvendra Bhattacharyya, Mainak Bose, Yogaditya Chakrabarty, Bahnisikha Barman, and Susanta Chatterjee

DOI: <https://doi.org/10.26508/lsa.201800161>

Corresponding author(s): Suvendra Bhattacharyya, CSIR-Indian Institute of Chemical Biology

Review Timeline:

Submission Date:	2018-08-15
Editorial Decision:	2018-09-18
Revision Received:	2019-12-07
Editorial Decision:	2020-01-10
Revision Received:	2020-01-15
Accepted:	2020-01-15

Scientific Editor: Andrea Leibfried

Transaction Report:

September 18, 2018

Re: Life Science Alliance manuscript #LSA-2018-00161-T

Dr. Suvendra N Bhattacharyya
CSIR-Indian Institute of Chemical Biology
Molecular and Human Genetics Division
RNA Biology Research Laboratory, 4 Raja S C Mullick Road
Kolkata, West Bengal 700032
India

Dear Dr. Bhattacharyya,

Thank you for submitting your manuscript entitled "Intracellular Trafficking of Argonaute 2 Acts as a Rate-Limiting Step for de novo miRNP Formation on Endoplasmic Reticulum Attached Polysomes in Mammalian Cells". The manuscript has been evaluated by expert reviewers, whose reports are appended below. Unfortunately, after an assessment of the reviewer feedback, our editorial decision is against publication in Life Science Alliance.

As you will see, the reviewers appreciate the interesting conclusions you put forward. However, both reviewers also note that the data cannot be properly evaluated and that many issues would need to get addressed to support your conclusions.

Although your manuscript is intriguing, we feel that the points raised by the reviewers are more substantial than can be addressed in a typical revision period and that furthermore the outcome of a revision is rather unclear at this stage. We therefore concluded that we have to return your manuscript to you with the message that we cannot publish it here. We are very sorry that we cannot bring better news, and we hope that the comments below will prove constructive as your work progresses.

Thank you for thinking of Life Science Alliance as an appropriate place to publish your work. Given the interest in the topic, we would be open to resubmission to Life Science Alliance of a significantly revised and extended manuscript that fully addresses the reviewers' concerns and is subject to further peer-review in the future and would need strong support from the reviewers. If you would like to resubmit this work to Life Science Alliance in the future, please contact the journal office to discuss an appeal of this decision or you may submit an appeal directly through our manuscript submission system. Please note that priority and novelty would be reassessed at resubmission.

Sincerely,

Andrea Leibfried, PhD
Executive Editor
Life Science Alliance
Meyerohofstr. 1
69117 Heidelberg, Germany
t +49 6221 8891 502
e a.leibfried@life-science-alliance.org

Reviewer #1 (Comments to the Authors (Required)):

Review of Bose et al. manuscript

This study reported the association of newly synthesized miR-122 with ER and rER in the presence of its target RNA. It was found that de-tethering of mitochondria from ER led to accumulation of Dicer and Ago2 in endosome fractions and a corresponding decrease in the association of miR-122 with Ago 2 and rER. In vitro assays suggested that the "transfer" of Ago2 from endosomes to ER requires mitochondria.

The cell biology of RNAi deserves attention, and from this perspective, the authors are commendable for exploring an under-studied area in RNAi. The work could be informative to the field of RNAi. However, the manuscript was very poorly written, such that it is very hard to read. Many instances of over interpretation or incorrect wording exists. Professional language editing is necessary, but language is not the only issue in clarity. Figure legends in general lack specific information necessary to understand how the experiments were done.

Figure 1A: need to show original data that led to the graph; when comparing the three fractions, how was normalization done? Against AGO2 levels? There was not enough description in the figure legend or in Methods. This is a huge problem throughout the manuscript.

Figure 1C: should talk about how "% specific activity" was calculated.

Figure 1D: need to show original data for this graph. How was normalization done? How was "microsome" compared to "total"?

Figure 1E: need a no-treatment control; need to show original data

Statement: "The de novo synthesized miR-122, accumulated from an inducible expression construct in HEK293 cells, along with its target mRNA RL-3xbulge-miR-122 was predominantly co-isolated with ribosomes in the supernatant after KCl-Puromycin treatment of rER (Fig 1E)".

The miR-122 and the target mRNA were in the supernatant, but the authors did not show that they were together with ribosomes in the supernatant. They need to detect ribosomes in the western blots. Even if ribosomes are in the supernatant, the word "co-isolate" implies association, which is not shown.

Figure 2A: The text that cited this figure mentions "increased miR-122". What is "increased miR-122"? Where is it in this figure? Presumably, expression of miR-122 was induced, but the figure legend did not mention how long after induction samples were taken, and what was the level of miR-122 before induction.

The U6 snRNA is not a good internal control for this experiment. The authors should consider other RNAs, for instance tRNAs.

Figure 2C: according to the text in Results, this should be about newly synthesized miR-122 upon induction with doxycyclin for 24 hours. So presumably the plot should be the levels of the miRNA at

24 hr after induction. It would be nice to know how fast the miRNA was made. Does the 24-hr time point still reflect "newly" synthesized miR-122?

Figure 2D: need to show original data behind the plot. Was it northern blot or RT-PCR? What was used as the normalization control? For the western blots, what is the relative amount of microsome relative to "total" that was used? For example, if they isolated microsome from 100 ul of total extracts and used all of the microsome fraction in the western blot while only using 10 ul of total extracts, then the two lanes cannot be directly compared. I wonder whether one can really reach the conclusion that AGO2 is "enriched" in rER.

Figure 2E: miR-122 was increasing in amount over time. Why does the text say "after 12h or miR-122 induction, a steep increase ..." (page 8)

Figure 2G: need to show original data leading to this graph in supplemental data.

Figure 3A and related experiments in Figure 3: There was not enough description of the similarity and/or different between "digitonin-insoluble membrane" and "microsomes/rER".

Figure 3B: target RNA associated with membrane more when pre-miR122 was present. The authors did not comment on this observation. The authors' previous studies found that the target RNA of miR-122 enhances the biogenesis of miR-122 and the target RNA's association with membrane precedes the association of miR-122 with rER. But here, the miRNA seems to affect the membrane association of the target RNA. It is not consistent with their previous observation.

"It strongly suggests target driven mature miRNA formation on ER-attached polysomes and mRNA targeting to polysomes to precede the de novo miRNP formation (Fig 3F)."

How does Figure 3F lead to this conclusion? Figure 3F only shows that the target RNA is associated with polysomes prior to miR-122 being associated with polysomes. No data on miRNP was included in the figure.

Figure 5A: not cited in text

Figure 6H: formatting issue

Second paragraph of Discussion: Ago2-uncoupled miRNAs - do you mean "miRNA-uncoupled Ago2"?

"Although a fraction of the Ago2 may, as a part of quality control check, undergo degradation upon targeting of Ago2 to lysosomal compartment (Bose et al, 2017; Gibbins et al, 2012), a large fraction of the same is available and get trafficked back to ER for its loading with new mature miRNAs processed by associated Dicer1 on the polysomes attached with the ER"

What was the evidence for the conclusion that a large fraction of Ago2 is trafficked back to ER for loading new miRNAs?

Page11: the authors did not introduce FH-Ucp2 before describing the related results. There was no description of it in Introduction either.

The Mfn depletion mutant is actually an RNAi line, the authors should use a knockout line.

The protein localization analyses in Figure 6C: the authors did not provide any negative controls for the immunofluorescence. Also, the authors should try to directly observe the YFP and DsRed signals rather than doing immunofluorescence.

Reviewer #2 (Comments to the Authors (Required)):

This paper by Bose et al. follows a series of recent papers dissecting localization of miRNA complexes over the course of pre-miRNA biogenesis to target recognition and degradation. There is an impressive amount of work in the paper and the authors have often used multiple approaches to make their points by using in vitro assays and cell fractionations of different types. I found the paper very challenging to evaluate because many times important controls were lacking, or how measurements were made or quantified were lacking. While I believe the research has a good foundation, publication would require addressing many questions about methods, adding controls to several experiments and moderating conclusions particularly about the strength of data on rough ER or polysome association of processes.

Specific Comments

Fig.1A - How much Ago2 is IP'ed in each fraction. How do you know that you IP all the Ago2 or that IP isn't preferential for one type of Ago2 complex. Could it be that you just IP more Ago2 in membrane fractions and that explains everything. Required to add western blot of total fractions before and after IP, as well as IP.

Fig.1B. Hrs marker of early endosome is in fraction 1. No markers of organelles peak in fractions containing Dicer and Ago2 so cannot conclude what these organelles are. Marker for late endosomes or other is required that peaks at fraction 3.

Fig.1C Error bars on graph of miRISC-122 activity? Single experiment?

Why does pre-miR-122 peak in fraction 8 where in Fig1B none of the components peak: Dicer, Ago2 are not abundant, nor any of the organelle markers. What organelle is this? If gradient is different should show blots.

Fig.1D-E. Microsomes are a rough fractionation that likely also contain many small vesicles from sources other than ER. Cannot conclude this is rough ER unless show results for both + and - KCl-puromycin on same gels and RT-qPCR quantification. Indeed, 500 mM KCl will remove many things from many membranes, so effects specific to puromycin or another reagent may be required to pin this to rough ER with ribosomes vs. other contaminating membranes. The authors should provide evidence that these fractions lack markers of early and late endosomes, or endosomal trafficking vesicles to support their claim.

Fig.2A - stats?, At what time point were these done missing from Figure legend

Fig.2A and B. Need to show blots for Ago2 IPs so can evaluate whether these effects are only due to variations in how much Ago2 is IP'ed between samples.

Evaluate throughout paper, what is actually depicted in graph. For example in Fig.2D is miR-122 normalized to something? If it is normalized to let-7 that could be a major complication in interpretation. This is a consistent problem throughout the paper and makes it impossible to properly evaluate the data. Same issue in Fig.2B, C

Details on many methods are missing: mitochondrial fractionation, rabbit reticulocyte assay source. No validation of mitochondrial fractionation and purity from rER or endosomes provided.

All of Fig.2 - the only differences in the time course that are significant is at 25 h so cannot conclude about kinetics.

Fig.2C - How can we conclude that inducible/de novo miR-122 distributes differently than steady-state miR-122 if the steady-state data is not shown? I realize it may not be expressed in HEK293, but this prevents a conclusion.

Fig.2E-H. If the point is to suggest what happens when a target mRNA is present then data with and without target mRNA need to be shown. This is missing in particular for polysome data.

Fig.3 How do the authors know for certain that they are measuring mature miR-122 and not pre-miR-122 as well?

Fig.3F The authors conclusions about the kinetics of recruitment of miR-122 vs. target mRNA are not supported by the data. The only significant difference is in miR-122 at early time points - which contradicts the authors proposal that the target mRNA is recruited first. In fact data in Fig2E would suggest that miRNA causes target recruitment and recruitment improves miRNA recruitment.

Fig.4 Provide data confirming expression and knockdown of Ucp2 and Mfn, as well as images of mitochondria in these cells to demonstrate expected phenotypes.

Fig4D - Why normalize to EE values of Ago2 associated miR-122? Then should show that these don't change and account for difference in ER values. Again, how do you know that this is not due to variation in amounts of Ago2 IP'ed either. Need to show western blots of IPs, ideally quantified over multiple experiments.

EV1D. Effects on Dicer1 co-localization are not apparent. Where are the statistical analyses? In Fig.6B change in Dicer could be in quantity detected, not overall distribution shift to EE/MVB. Is this important for the conclusions of the paper?

The blot for Ago2 in Fig.6F is very difficult to interpret and to my eyes looks like the opposite of the result suggested by the authors.

In general connections of mitochondria to ER are unlikely to be established in the same way in in vitro systems as they are in vivo. Same for interactions of MVB with ER. These caveats or evidence to the contrary should be included in the manuscript as this questions most of the in vitro assays.

MFn and Ucp have multiple effects on cells due their fundamental role in cellular metabolism and energy. It is challenging to attribute effects exclusively to ER-mitochondria contacts. This should be noted by the authors more emphatically.

Providing luciferase or target protein level assays in Mfn^{-/-} cells would improve conclusions on effects on miRNA activity.

Language should be revised. A few quirks like parallely and juxta positioning as two words, or wrong verb tenses.

Response to Reviewer's Comments:

[Reviewer's comments are in bold]

Reviewer #1

The cell biology of RNAi deserves attention, and from this perspective, the authors are commendable for exploring an under-studied area in RNAi. The work could be informative to the field of RNAi. However, the manuscript was very poorly written, such that it is very hard to read. Many instances of over interpretation or incorrect wording exists. Professional language editing is necessary, but language is not the only issue in clarity. Figure legends in general lack specific information necessary to understand how the experiments were done.

We appreciate the reviewer concern and apologise for the incorrect wording, language quality and lack of clarity that were obscuring the scientific merit of the manuscript in its previous version. Now it has been improved. We have tried to bring more clarity both in the main text and figure legends to increase the language quality and throughout the process also have done necessary experiments to strengthen our claim.

Figure 1A: need to show original data that led to the graph; when comparing the three fractions, how was normalization done? Against AGO2 levels? There was not enough description in the figure legend or in Methods. This is a huge problem throughout the manuscript.

We do apologies for the inconveniences and confusion. We have now tried to rectify and modify the manuscript as suggested by both the reviewers. We have incorporated one set of original data use for quantification purpose in 1A. Normalization has been done against the amount of Ago2 immunoprecipitated from each fraction and the procedure has been elaborately described in Material and Methods section under the sub-heading "In vitro RISC Cleavage assay and Data analysis". Fig 1A legend has also been modified to add the clarity in the description of the figure panel.

Figure 1C: should talk about how "% specific activity" was calculated.

Like many other queries, this query raised by reviewer is also justified. We have tried to rectify the lacunae in our manuscript. Detailed procedure for calculation of "% specific activity" has been added "Material and Methods" sections under sub-heading "In vitro RISC Cleavage assay and Data analysis". The calculation has been also shown in Source Data for Fig.1.

Figure 1D: need to show original data for this graph. How was normalization done? How was "microsomes" compared to "total"?

We have tried to clarify the normalization procedure for "microsomes" compared to "total". We have modified and added a paragraph on "Cell Fractionations" sub-section under "Materials and Methods" section to clarify the normalization procedure for each organelle fractions against the total off all fractions. Figure legends also rectified to bring clarity. Original data has been shown in Source Data for Fig.1.

Figure 1E: need a no-treatment control; need to show original data

We have included a "no-treatment control" to address the concern raised by reviewer and have included it in the Figure 1F. The original data has been shown in Source Data for Fig.1E

Statement: "The de novo synthesized miR-122, accumulated from an inducible expression construct in HEK293 cells, along with its target mRNA RL-3xbulge-miR-122 was predominantly co-isolated with ribosomes in the supernatant after KCl-Puromycin treatment of rER (Fig 1E)". The miR-122 and the target mRNA were in the supernatant, but the authors did not show that they were together with ribosomes in the supernatant. They need to detect ribosomes in the western blots. Even if ribosomes are in the supernatant, the word "co-isolate" implies association, which is not shown.

This is an important concern. We have tried to address this issue with addition of necessary figure panels and re-structuring few parts of "Results" to strengthen the conclusion but without doing an over-statement. We have added a panel in Fig 1F to show that there was no leaching of ribosomes (shown by L3 as marker protein) in absence of Puromycin-KCl and ribosomes and Ago2 were co-extracted upon KCl-Puromycin treatment. We have also introduced additional control for the same (please see the panel below)

Figure 1. Specificity of KCl-Puromycin based extraction of ribosomes from microsome. Extraction was done with or without KCl to show absence of ribosomal marker protein in supernatant without KCl treatment. Calnexin and L7a were used as markers of rough ER and extracted ribosome respectively. Puromycin was present on both the extraction buffers.

.Figure 2A: The text that cited this figure mentions "increased miR-122". What is "increased miR-122"? Where is it in this figure? Presumably, expression of miR-122 was induced, but the figure legend did not mention how long after induction samples were taken, and what was the level of miR-122 before induction. The U6 snRNA is not a good internal control for this experiment. The authors should consider other RNAs, for instance tRNAs.

Probably there were some miscommunications that cause the confusion while connecting the observations described in the figure with result section in main text.

There was no miR-122 induction in this experiment. HEK 293 cells were co-transfected with miR-122 expression plasmid that drive expression of miR-122 precursor from a constitutive U6 promoter (Bose and Bhattacharyya, 2016) along with a target mRNA expressing plasmid driven by CMV promoter. Fractionation was done with Digitonin and that was followed by RNA isolation from each fractions. RNA (equal amounts) was subjected to Northern blotting to score the mature miR-122 present there. The "increased miR-122" refers to the increased level of mature miR-122 detected specifically in the membrane fraction (also quantified in the adjacent graph). The northern blot for total miR-122 had been cropped in the main figure; the whole blot is now provided in Original Data Fig.2 file along with total RNA stained with Ethidium Bromide. Main text section under "Target dependent miRNA biogenesis happens on rER" has also been modified. Quantifications for total and other fractions have been done from more than one northern blot. Picture of one such representative blot has been shown.

Using tRNA as an internal control would be a good suggestion. However, we are not comparing between cytosol and membrane fractions in this experiment. Where, the varying amounts of U6 between the 2 fractions were expected. But comparison between membranous fractions with or without target mRNA is doable with U6 as the amount of U6 present in each case should be similar and also were corroborated by Ct values obtained for U6 in qRT-PCR analysis (See Original data for Figure 2A). Using the U6 value as reference, significant amount of increase of mature miR-122 level attached with the membrane fraction in presence of target message is

visible. The digitonin fractionation protocol followed both for RL-con and RL3xbulge-miR-122 sets were identical and were confirmed by similar extent of U6 RNA present in cytoplasmic fractions in both conditions. Pre-miR-122 was transcribed using a U6 promoter in HEK 293 cells where this miRNA was expressed in a heterologous context. Therefore, U6 would be a better choice as an internal control to rule out any change in transcription rate of pre-miR-122. The RNA profile shown in Fig2A also suggests that the bottom-most tRNA content also don't change between experimental conditions. Also, the northern blot was done with equivalent amounts of RNA which ensures that the change observed between miR-122 levels between the membrane fractions of RL-con and RL-3xbulgemir-122 sets has been valid.

Figure 2C: according to the text in Results, this should be about newly synthesized miR-122 upon induction with doxycycline for 24 hours. So presumably the plot should be the levels of the miRNA at 24 hr after induction. It would be nice to know how fast the miRNA was made. Does the 24-hr time point still reflect "newly" synthesized miR-122?

miRNAs are stable molecules and from kinetics of miR-122 induction it seems there has been a lag phase upto 4h then onwards there has been a sharp increase in inducible miR-122 expression. The increase miRNAs further increased to till 24h time we followed and possibly increased further. Fig. 2C reflects the enrichment of cognate miRNA on ER fraction in presence of target mRNA. We have checked miRNA compartmentalization on ER at early time-points after doxycycline treatment to check "how first miRNA was made and compartmentalized" (Fig EV1).

The 24h time point for the analysis was chosen to let the miRNA synthesis to reach an dynamic equilibrium and also to allow significant miRNA to be present in each fraction to get the detection limit in reliable ranges (i.e. in MVB fractions). Our aim was to investigate the fate of miRNP complexes on rER after repression at a time when ample amount of miRNPs are present in this compartment. Such that we could monitor the further trafficking processes. We agree with the point that we can't technically call the miRNPs as "New" but as the miR-122 is absent in HEK 293 cells and upon induction majority of miR-122 formed at the early time points are primarily with rER, we conclude at 24 h even when the system possibly reached an equilibrium the majority of miR-122 formed over the 24h induction time are with ER fraction. We have change the text and refined our claim to state that induced miR-122 found to be enriched with rER fraction rather than the newly synthesized miRNAs.

Figure 2D: need to show original data behind the plot. Was it northern blot or RT-PCR? What was used as the normalization control? For the western blots, what is the relative amount of microsomes relative to "total" that was used? For example, if they isolated microsomes from 100 ul of total extracts and used all of the microsomes fraction in the western blot while only using 10 ul of total extracts, then the two lanes cannot be directly compared. I wonder whether one can really reach the conclusion that AGO2 is "enriched" in rER.

We are thankful to the reviewer for pointing out such missing information that should have been addressed in manuscript. We have agreed on this part and modified the text and legends accordingly.

Fig. 2D original real-time PCR data has been provided as Original Source data for Figure 2.

Fig. 2D data has not been originated from Northern Blot, rather it was a qRT-PCR data. We have used U6 snRNA as endogenous housekeeping target. To understand target mRNA dependent miRNA biogenesis and compartmentalization; we have normalized level of RL-3xB-miR-122, a reporter target mRNA having miR-122 binding sites on its' 3' UTR, with level of RL-3xB-let-7a, a reporter with non-miR-122 binding sites on its' 3' UTR in a miR-122 inducible system.

Probably here reviewer is referring to Fig. 1D. We have clarified the same concern about normalization done in Fig. 1D also. We didn't compare total vs. microsomes on cell equivalent amount to avoid such above mentioned discrepancies. Even every other time microsomes yield from same starting number of cells may differ. To avoid this, protein equivalent amount of both the total and microsomal fractions had been used to detect protein enrichment in total vs microsomes on western blots. We have modified the text and have added a paragraph on "Cell Fractionations" sections under "Materials and Methods" to clarify the normalization procedure.

Figure 2E: miR-122 was increasing in amount over time. Why does the text say "after 12h or miR-122 induction, a steep increase ..." (page 8)

Here, we agree with the reviewer that we should have shown miR-122 synthesis pattern at earlier time-point also to comment on the time-point from which miR-122 synthesis shows sharp increase. To address this issue, we have checked miR-122 synthesis at earlier time-points also and have shown it in Fig EV1.

We did not see “steep” increase in miR-122 formation at early time-points (i.e. 2h. and 4h.) after doxycycline treatment. After 4h of induction, we could see significant increase of miR-122 synthesis. After this observation, we have also modified our text under ‘Kinetics of compartmentalization of miRNAs synthesized in presence of target mRNA’ on the results section.

Figure 2G: need to show original data leading to this graph in supplemental data.

We have incorporated the above mentioned original file in our revised manuscript. Please see Source data Fig. 2 file.

Figure 3A and related experiments in Figure 3: There was not enough description of the similarity and/or difference between "digitonin-insoluble membrane" and "microsomes/rER".

The point raised by the referee is important and we are agreeing on this. We have included a brief understanding on this issue at the starting of ‘Kinetics of compartmentalization of miRNAs synthesized in presence of target mRNA’ section on the results part.

Figure 3B: target RNA associated with membrane more when pre-miR122 was present. The authors did not comment on this observation. The authors' previous studies found that the target RNA of miR-122 enhances the biogenesis of miR-122 and the target RNA's association with membrane precedes the association of miR-122 with rER. But here, the miRNA seems to affect the membrane association of the target RNA. It is not consistent with their previous observation.

Although, there seems to be an apparent contradiction but careful examination of the context explains both results. Target RNA binding with rER/polysomes precedes miRNA binding. The step of miRNP interaction with mRNA should increase miRNP biogenesis and in steady state it will help further retention of messages to rER that are bound to the *de novo* formed miRNPs. Therefore a *feed-forward reaction* both to enhance mRNA compartmentalization to the membrane associated pool and miRNA biogenesis should occur simultaneously. We have introduced the clarification in the respective text part of the result section to avoid such confusion.

Therefore the whole event can be classified into two steps:

a. In presence of cognate miRNAs, target mRNA would have more chance to get sequester on membrane in steady state or initial time (Fig 3B).

b. On the 2nd step, after target mRNA-miRNA binding on membrane, target mRNA would increase processivity of miRNP associated Dicer that would ultimately lead to enhanced target-dependent miRNA biogenesis. [Bose et al. 2016](Fig. 3F)

Biogenesis would lead to enhanced association of miRNA on rER, more recruitment of target mRNA and so on in an cooperative feed forward reaction.

"It strongly suggests target driven mature miRNA formation on ER-attached polysomes and mRNA targeting to polysomes to precede the de novo miRNP formation (Fig 3F)."

How does Figure 3F lead to this conclusion? Figure 3F only shows that the target RNA is associated with polysomes prior to miR-122 being associated with polysomes. No data on miRNP was included in the figure.

The reviewer is correct to point this out. Yes; although it can be assumed that mature miRNAs formed during the reaction should be part of the miRNPs as the target driven biogenesis is known to affect the post miRNA processing step (Bose and Bhattacharyya, 2016), it is technically not possible to conclude on miRNPs status from this experiment. We have modified our claim accordingly.

Figure 5A: not cited in text

We are sorry for this unintentional fault. We have incorporated the necessary citation in the manuscript.

Figure 6H: formatting issue

We have resolved this issue in the current version. Most probably this was at the time of pdf conversion.

Second paragraph of Discussion: Ago2-uncoupled miRNAs - do you mean "miRNA-uncoupled Ago2"?

In the context that is referred here, it is the miRNAs that get uncoupled from Ago2 but it also creates pool of Ago2 without miRNAs.

"Although a fraction of the Ago2 may, as a part of quality control check, undergo degradation upon targeting of Ago2 to lysosomal compartment (Bose et al, 2017; Gibbings et al, 2012), a large fraction of the same is available and get trafficked back to ER for its loading with new mature miRNAs processed by associated Dicer1 on the polysomes attached with the ER"

What was the evidence for the conclusion that a large fraction of Ago2 is trafficked back to ER for loading new miRNAs?

The point raised by the reviewer is one of the important observations of our manuscript. Also, it was quite critical to prove this phenomenon quantitatively under steady organelle dynamics condition. Hence, we chose a system where organelle dynamics could be hampered (like in *Mfn2*^{-/-} cells) and see the differential effect. Figs 6F and 6G depicts an "in vitro inter-organelle Ago2 transfer assay" where Ago2 recycling to rER could be seen in a process that has been hampered when mitochondria tethering defective microsomes has been incubated with normal endosomes.

However, it would be incorrect to claim that a large fraction of Ago2 associated with MVB should be trafficked in a unit time for *de novo* miRNP formation. It may be influenced by several factors including mitochondrial activity but as we could not comment on net amount of Ago2 trafficked for each round of mRNP formation we have rewritten the phase to be factually correct with the statement.

Page11: the authors did not introduce FH-Ucp2 before describing the related results. There was no description of it in Introduction either.

As per necessary modifications asked by reviewer we have introduced accordingly.

We have added the requisite description and rationale of use of FH-Ucp2 on page 11 before describing the connected results. Also we have inducted a brief introduction of Ucp2 on page 4, in Introduction.

The Mfn2 depletion mutant is actually an RNAi line, the authors should use a knockout line.

We have used the MEF cells from *Mfn2*^{-/-} knock-out mice for our studies wherever indicated.

The protein localization analyses in Figure 6C: the authors did not provide any negative controls for the immunofluorescence.

Regarding this comment we are bit confused as we don't fully understand reviewers concern regarding "negative controls". As far as our understanding, it may be difficult to visualize the colocalization in merged images, so we have included panels for the microscopic analysis showing representative ROIs of individual channels in addition to the merged channel image as we did also for Figure EV3 also. This was done to remove the ambiguity between the images to clearly show the variation as observed and quantified in colocalization data in Fig. 6B.

Also, the authors should try to directly observe the YFP and DsRed signals rather than doing immunofluorescence.

We regret for the confusion caused. The fluorescence from endosome targeted YFP (Endo-YFP) and ER targeted DsRed (ER-DsRed) had been directly observed as have also mentioned in the respective figure legends. Immunofluorescence has only been used for either Ago2 (Fig. 6A) or Dicer (Fig. EV3D) protein detection in PFA fixed cells transiently expressing Endo-YFP or ER-DsRed. We have also included clarification regarding the same in text and materials and Method part to avoid any further confusion.

Response to **Reviewer2's** Comments:

I found the paper very challenging to evaluate because many times important controls were lacking, or how measurements were made or quantified were lacking. While I believe the research has a good foundation, publication would require addressing many questions about methods, adding controls to several experiments and moderating conclusions particularly about the strength of data on rough ER or polysome association of processes.

We are happy that the reviewer has liked the concept of this manuscript. The concerns he has expressed are appropriate and by addressing these concerns, we have tried to get the revised manuscript improved. It now comprises with high quality data with all necessary controls and explanations of their use. We have also moderated the conclusions drawn based on the clarity and strength of the data in this revised version of the manuscript.

Fig.1A - How much Ago2 is IP'ed in each fraction. How do you know that you IP all the Ago2 or that IP isn't preferential for one type of Ago2 complex. Could it be that you just IP more Ago2 in membrane fractions and that explains everything. Required to add western blot of total fractions before and after IP, as well as IP.

The amount of FH-Ago2 IPed and amount of FH-Ago2 present in each fraction have now been shown as western blot data in Fig1A.

It may be noted that both total and specific activity of the membrane isolated FH-Ago2 is much higher in terms of miRNA bound and RISC cleavage activity as depicted in this panel. Similar observation has also been noted in panel 2A. In both cases, we could not detect Ago2 in the residual fraction after the IP by western blot. So no blot has been shown for that fraction.

Fig.1B. Hrs marker of early endosome is in fraction 1. No markers of organelles peak in fractions containing Dicer and Ago2 so cannot conclude what these organelles are. Marker for late endosomes or other is required that peaks at fraction 3.

The concern raised by the reviewer is valid. We have modified Fig 1B where we have included western blots for other endosomal marker proteins to make the conclusion stronger. Late endosomal marker protein, Rab7 western blot has been included herein.

Fig.1C Error bars on graph of miRISC-122 activity? Single experiment? Why does pre-miR-122 peak in fraction 8 where in Fig1B none of the components peak:

Dicer, Ago2 are not abundant, nor any of the organelle markers. What organelle is this? If gradient is different should show blots.

The Markers for ER and Endosomes (HRS) have now been shown and from the protein profile it is obvious the RISC activity and pre-miR-122 are primarily attached with ER fractions that span over 6-9th fraction. However, the 2-4 fractions, enriched for endosomes, are co-migrating on the density gradient with majority of Ago2.

On Fig 1C the 3-30% Iodixanol gradient (Opti-prep) has been used to calculate % specific activity of miRISC 122 and % total pre-miR-122 level in individual fractions. The experiments involve RISC cleavage assays done with immunoprecipitated Ago2 from each fraction collected with gradual increase in density from 3-30%. As we have a consistent and continuous pattern of RISC cleavage end products formation in individual assays, where fractions 8-10 have Ago2 enrichment along with rER membrane marker and polysome, we have technically done three independent reactions with variable amount of Ago2 in 8-10 fractions to get similar values of specific activity. Same holds true for 2-4th fractions enriched with Ago2 associated with endosomes. Thus we consider the single experimental set has multiple experiments and has been informative enough to conclude that the specific activity of ER/polysome attached Ago2 has relatively higher specific activity compared to Endosome/MVB associated Ago2. The fraction 8 is positive for COX4 the mitochondrial marker protein and also for ER marker calnexin. It is likely the fraction containing the mitochondria –associated membrane (MAM). Pre-miR-122 has also accumulated in this fraction before they get processed by Dicer1.

Fig.1D-E. Microsomes are rough fractionations that likely also contain many small vesicles from sources other than ER. Cannot conclude this is rough ER unless show results for both + and - KCl-puromycin on same gels and RT-qPCR quantification. Indeed, 500 mM KCl will remove many things from many membranes, so effects specific to puromycin or another reagent may be required to pin this to rough ER with ribosomes vs. other contaminating membranes. The authors should provide evidence that these fractions lack markers of early and late endosomes, or endosomal trafficking vesicles to support their claim.

On suggestion of the reviewer, we have done the extraction reaction also without KCl/puromycin and have shown the protein fractionation data and miRNA enrichment data in both cases. We have noted no enrichment of Ago2 and miRNA when extraction was done without KCl/puromycin (Fig. 1F). Puromycin is there to ensure premature run off of the polysomes and to get

them dissociated when treated. However, KCl alone may be enough to get the Ago2, miRNA and polysomes (undissociated to monosomes) all in the supernatant after the treatment. Therefore only KCl extraction experiments will not be that informative to conclude on exclusiveness of rough ER as source of miRNPs and therefore we preferred to opt out the fractionation with only KCl. The contamination from endosomal fractions in isolated microsomes is marginal (as shown by weak Hrs intensity in microsome derived fractions in Fig 1F) and that did not get extracted in soluble phase with KCl /puromycin. The purity on microsomes has been tested in other context in previous reports from our group (Bose et al. 2017). Please note the microsome data is only a supportive data of results presented in Figure 1A-D where more rigorous fractionation over an Optiprep gradient run in an isotonic condition followed by analysis of the content revealed the same findings of ER/polysomal enrichments of *de novo* miRNPs in mammalian cells. Because of existence of potential ambiguities with microsomes, we also did not rely exclusively on this data to conclude on miRNPs location in mammalian cells.

In support of ribosomal association of miRNA and target mRNA, we wish to refer two previous publications on that. Caudy *et al.* 2002 reported ribosomal association of miRNA along with Ago2 protein which is also absent in cytosolic fractions. Barman *et al.*, 2015 reported and validated polysomal enrichment of miRNPs as well as target mRNAs using various biochemical approaches. Two of these independent observations concomitantly strengthen the fact that miRNPs and target mRNA remain associated with polysome. Here, we have preceded our work based on those previous observations.

Barman *et al.*, 2015 also validated isolation of ribosomes from rER fractions using KCl/Puromycin based technique. Their observation suggests enrichment of target mRNA, miRNPs specifically with supernatant fractions (polysome) but not with pellet fractions (ER fractions).

Fig.2A - stats?, At what time point were these done missing from Figure legend.

Fig.2A and B. Need to show blots for Ago2 IPs so can evaluate whether these effects are only due to variations in how much Ago2 is IP'ed between samples.

The query raised by the reviewer is probably due to some misunderstanding of what we wanted to say. For Fig 2A, miR-122 was not induced in this experiment rather HEK 293 cells were co-transfected with miR-122 and target mRNA expressing plasmids, fractionated with Digitonin

after 48h and RNA isolated from the fractions (equal amounts) were subjected to Northern blotting. This information is now part of the legends.

Normalization of the values were done against the amount of Ago2 immunoprecipitated. For Fig 2A, amount of IPed FH-Ago2 used in RISC assay has been already there in the figure.

For 2B, the blot for FH-Ago2 immunoprecipitated material is added in Fig 2B.

Evaluate throughout paper, what is actually depicted in graph. For example in Fig.2D is miR-122 normalized to something? If it is normalized to let-7 that could be a major complication in interpretation. This is a consistent problem throughout the paper and makes it impossible to properly evaluate the data. Same issue in Fig.2B, C

The concern raised by the reviewer is understood and addressed herein. We have modified the language in Results and Methods sections for better understanding of normalization procedures.

On Fig. 2D, miR-122 level had been normalized with endogenous U6 snRNA and not against let-7a level. Here, RL-3xb-let-7a has been used as negative control as this reporter mRNA does not have miR-122 binding sites although it has same 3' UTR length compared to RL-3xb-miR-122. So, it can be hypothesized, reporter target mRNA that has its cognate miRNA binding site induces cellular miRNA biogenesis as well as its increased ER compartmentalization.

On Fig 2C, let-7a level has been monitored to see steady-state level distribution of endogenous miRNA on cellular fractions. Parallel to that, we wanted to follow the differential miRNA compartmentalization for newly synthesized miRNAs. To decipher this, cells were transfected with inducible exogenous miRNA expressing plasmid in presence of its' reporter target. Induced miRNA (i.e. miR-122) has been found to increase significantly with ER fractions. On Fig 2C, we observed subcellular compartmentalization of newly-formed miRNA. We had introduced a Tet-On system with inducible pre-miR-122 expressing plasmids where miR-122 don't otherwise not expressed in detectable level in HEK 293 cells. Normalization against U6 RNA was done in each case.

On Fig 2B, we have wanted to observe target-dependent increase in miRNA biogenesis and activity, happening with microsome fraction, by using RISC cleavage assay. There, RL-con has acted as negative control with no cognate-miRNA binding site against RL-3xb-miR-122 that has three miR-122 binding sites. We have observed, the presence of target mRNA with cognate

miRNA binding sites on 3'UTR, has increased miR-122 enrichment with microsomes [northern blot for miR-122]. RISC mediated cleavage assay of these samples also showed increased cleavage of substrate by FH-Ago2 immunoprecipitated from microsomes isolated from RL-3xb-miR-122 expressing cells. Equal amount of RNA were used for quantification purpose.

Details on many methods are missing: mitochondrial fractionation, rabbit reticulocyte assay source. No validation of mitochondrial fractionation and purity from rER or endosomes provided.

We apologies for the missing part of the methods that were been pointed out by the reviewer. We have inducted new parts and thoroughly updated “Materials and Methods” part. Methods for Crude Mitochondrial Fractionation have been included now in Materials and Methods section. Rabbit reticulocyte assay sources have been provided on page 20 under *In vitro* translation assay of Materials and Methods section.

Concern raised regarding rER purity check has been presented in Figure 1.

Here, our aim was to study on impact of mitochondria-ER dynamics on miRNP biogenesis, activity and its' trafficking. We didn't emphasize on isolation of mitochondria in pure from because the effect observed may be shown by the mitochondria associated ER, otherwise we could have followed a different method. Here, we wish to isolate mitochondria enriched fraction having ER tethered to it and we expect mitochondria associated membrane (MAM) should also an integral part of our "Crude mitochondria fraction". Respective western blots to confirm the mitochondria enriched fractions have been presented in Fig 5.

All of Fig.2 - the only differences in the time course that are significant is at 25 h so cannot conclude about kinetics.

The concern raised by reviewer is true and we have done earlier time points experiments also to strengthen the points on time kinetics described in EV1. As suggested by reviewer, new data has been incorporated in EV1 to follow the changes at initial time points.

Fig.2C - How can we conclude that inducible/de novo miR-122 distributes differently than steady-state miR-122 if the steady-state data is not shown? I realize it may not be expressed in HEK293, but this prevents a conclusion.

The concern raised by the reviewer is partly true. Although Fig 2C focuses on differential cognate miRNA distribution in presence of its' target mRNA. Let-7a, an endogenous miRNA

which have no binding site for miR-122 reporter target mRNA, has been implicated as control to observe steady-state distribution of pre-existing endogenous miRNAs. However, in experiments describes throughout Fig.1 all assays and distribution pattern of miR-122 steady state level in HEK293 cells and data from Fig. 1C and 1D support our conclusion that newly synthesized miRNA association has strong preference/exclusiveness for rER association.

Fig.2E-H. If the point is to suggest what happens when a target mRNA is present then data with and without target mRNA need to be shown. This is missing in particular for polysome data.

We have tried primarily to score what happening during miRNA biogenesis and as target RNA has a positive influence on miRNA biogenesis as reported earlier (Bose et al 2016), we have tried to use a condition where the target driven biogenesis would be maximum in presence of target mRNAs. However miRNA biogenesis is a wide spread event happening in mammalian cells and presence of all endogenous mRNAs that have potential target sites for the cognate miRNAs should positively influence the biogenesis steps. Therefore, it is unlikely that the mechanism of miRNA biogenesis would be different even in absence of exogeneously expressed reporter mRNAs. The presence of exogenous targets may only accelerate the process kinetically by enhancing the Dicer processivity. As previously from our lab, we have shown elaborately how target mRNA may drive biogenesis of its cognate miRNA, here we didn't want to reprove an already established phenomenon (Bose et al, 2016). Rather, our curiosity was where the target dependent biogenesis is happening. So, on those above referred figures our aim was to validate the site of target dependent biogenesis.

In reference to polysomal data on Fig 2F, we have revalidated those findings in Fig. 3 using *in vitro* system. Fig 3E specifically depicts about differential polysomal association of miR-122 influenced by its' reporter target mRNA.

However, to address the reviewer concern, we have done the polysomal experiments and have obtained the data as per expectation and this is shown below:.

Figure II. Effect of expression of target mRNA on polysome association of miR-122. White bar represents value in RL-control and black bar represents value in RL-3x bulge-miR-122 expressing cells.

Fig.3 how do the authors know for certain that they are measuring mature miR-122 and not pre-miR-122 as well?

We have used Taqman based miRNA assay system for detection of mature miRNA levels throughout the paper. Taqman based primers selectively amplifies mature miRNAs only but not the precursor-miRNA due to a stem-loop structure of primers used, which enables them only to bind and form cDNA from mature only.

Reference: <http://tools.thermofisher.com/content/sfs/brochures/taqman-microrna-assay-faqs.pdf>

Additionally we have performed the northern blot data for mature miRNAs in Fig1C, 2A and B where from the position of the band we could be sure that we are only following mature miRNAs.

Fig.3F The authors conclusions about the kinetics of recruitment of miR-122 vs. target mRNA are not supported by the data. The only significant difference is in miR-122 at early time points - which contradicts the authors proposal that the target mRNA is recruited first. In fact data in Fig2E would suggest that miRNA causes target recruitment and recruitment improves miRNA recruitment.

The concern raised by the author is true. We have rechecked the annotation and clearly mention it in the recent version. The data presented is consistent with our claim that target mRNAs recruited to the membrane that preceded the polysome association before the miRNAs get enriched there. Please also consider our reply concern raised by Referee 1 on similar point.

Fig.4 Provide data confirming expression and knockdown of Ucp2 and Mfn, as well as images of mitochondria in these cells to demonstrate expected phenotypes.

As per suggestions by the reviewer, we have added new blots showing expression of HA-Ucp2 and reduction off Mfn2 upon siMfn2 treatment which have been incorporated in Fig. EV2. The absence of Mfn2 in MEFs knock-out for Mfn2 was verified in our previous papers where we have used this cell for the first time (Chakraborty, et al. 2017). Microscopic images showing differential morphology of the mitochondria have also been inducted in Fig EV2.

Fig4D - Why normalize to EE values of Ago2 associated miR-122? Then should show that these don't change and account for difference in ER values. Again, how do you know that

this is not due to variation in amounts of Ago2 IP'ed either. Need to show western blots of IPs, ideally quantified over multiple experiments.

Respective IP-blots has been represented as Original data for Fig 4.

We are thankful to reviewer 2 for his critical review thus, presenting us with the chance to improve upon the clarity of our observation. To address the above points we have taken the reviewer's suggestion into account and included the level of Ago2 associated miR-122 levels in EE fraction. Where they have remained largely unchanged and thus helped us to use them for depicting the changes occurring in the ER fraction. Additionally we have mentioned the point in the revised version. Moreover as the reviewer has correctly pointed out that the changes in ER associated Ago2 bound miR-122 levels will alter depending upon the IPed Ago2 levels. Therefore, we have now included representative IPed Ago2 blots for Fig 4D. In these blots although the level of Ago2 IPed varied between the EE and ER fraction as per expectation due to the large variation in the associated Ago2 pools with these fractions (Fig 6B). Nevertheless, the variation in the IPed Ago2 levels between control and experimental sets of EE or ER are marginal. To account for even small variation in the IPed Ago2 levels we did densitometric quantification of these bands and used them to normalize the respective Ago2 associated miR-122 levels.

EV1D. Effects on Dicer1 co-localization are not apparent. Where are the statistical analyses? In Fig.6B change in Dicer could be in quantity detected, not overall distribution shift to EE/MVB. Is this important for the conclusions of the paper?

We appreciate the critical view of reviewer. We have tried to redo the same experiment but was unable to get any statistically significant difference in Ago2-Dicer1 interaction on EE in wild type and knock out samples. Figure EV1D thus was not imperative for supporting the conclusions made in this manuscript and we opt for not showing any data on Ago2-Dicer1 interaction change at subcellular level. However the data for total decrease in Ago2 Dicer1 interaction is within statistical significance and part of our main conclusion.

The Figure 6B (Now Figure EV 3A) is there to strengthen the observation of an overall distribution shift of Dicer to EE/MVB. All the Iodixanol gradient preparations were loaded with post nuclear supernatant volume containing equal protein (Estimated by Bradford Assay) prior to density based separation by ultracentrifugation. Moreover, the SDS-PAGE was done together on the same gel such that variations due to immuno-blotting and detection could be negated; the two panels i.e. left (WT) and right (Mfn2 mutant) were separately presented only for

aesthetic purpose. Any post imaging changes (brightness/contrast) of the blot panels were done together (Figure EV3). Furthermore, to compare any results between 2 panels it becomes imperative that it shall be normalized before making any observations or comments. We have introduced densitometrically quantified relative Dicer levels present in each fraction. Individual Dicer band intensities have been normalized against total Dicer band intensity over 10 fractions. It is clear from these observations that there is indeed an overall distribution shift of Dicer, Ago2 and Hsp70 to EE/MVB in our studies with Mfn2 negative MEFs. We are thankful to the reviewer for pointing out this ambiguity and hence providing us with a chance to improve the clarity of our observations.

The blot for Ago2 in Fig.6F is very difficult to interpret and to my eyes looks like the opposite of the result suggested by the authors.

The microscopic techniques to validate differential MVB to ER Ago2 trafficking in WT vs Mfn2^{-/-} MEFs could not be achieved as potential influence of Ago2 localized to other compartments would obscure the data interpretation in a quantitative manner. Although it was very challenging but we had to adopt a biochemical *in vitro* assay to see the changes. Absence of other contaminating fractions in the *in vitro* assays helped us to get a cleaner context to interpret the data. We could observe increased recycled Ago2 level on WT MEF microsomes compared to microsomes from Mfn2^{-/-} MEFs, even when re-isolated microsome levels were less for WT MEFs (i.e. less Calnexin level on WT MEFs) after re-isolation of microsomes. The weak band intensity suggests that only a small amount of Ago2 is expected to be transferred. Densitometry quantification was followed to measure the changes. The FH-Ago2 was not expressed in cells from where the microsomes were isolated and therefore all the FH-Ago2 recovered with re-isolated microsomes are from transferred Ago2 itself. The Ago2 associated with wild type cell microsomes after re-isolation has been in higher amount than Mfn2^{-/-} microsomes where no band of Ago2 has been detected (Middle column/uppermost panel).

In general connections of mitochondria to ER are unlikely to be established in the same way in in vitro systems as they are in vivo. Same for interactions of MVB with ER. These caveats or evidence to the contrary should be included in the manuscript as this questions most of the in vitro assays.

The concern raised by the author is very important one. We had to adopt *in vitro* systems in some cases where *in vivo* validations were not possible mostly due to technical limitations. On Fig. 3, we have re-validated compartmentalization of target mRNA dependent miRNA

biogenesis in the *in vitro* context. Whereas in Fig 2, we have validated most of our hypothesis with simultaneous experiments that are done primarily in cell lines (*in vivo*).

On Fig. 5E-F, we have adopted *in vitro* assay system to strengthen and revalidate our data obtained from Fig 5A-D. Reviewer's concern regarding inter-organelle communication/interaction in the *in vitro* system is however very much legitimate and that was the reason for us to strengthen our observation biochemically, microscopically or by *in vivo* experiments. In some cases, particularly in 5E-F and 6C-E, we had adopted *in vitro* translation system where we wanted to precisely decipher the exact organelle or mechanisms which leads to differential cellular trafficking. However, we have noted the existence of possible alternative derivation of *in vitro* and *in vivo* data and have pointed that out in appropriate places in the main text.

MFn and Ucp have multiple effects on cells due their fundamental role in cellular metabolism and energy. It is challenging to attribute effects exclusively to ER-mitochondria contacts. This should be noted by the authors more emphatically.

Yes, the concern raised by the author is very much appropriate. Previous observations from our Lab., (Chakraborty et al. 2017) suggested Ucp2 overexpression or loss of Mfn2 protein shows defective ER–endosome colocalization as well as changes in miRNP levels and stability. Mitochondrial depolarization also affects turnover of existing miRNPs by preventing their targeting to EE/MVBs, thus Ago2 trafficking would be expected to be significantly hampered. It is expected mitochondria as an energy reservoir imparts a huge role in ER functioning. Mito-ER tethering facilitates dynamicity of organelles that ultimately keeps cellular environment in steady-state conditions. We have mentioned this concern in the discussion part of the manuscript. We could not rule out the effect we observed on miRNP biogenesis due to mitochondrial dysfunction in cells with depolarized mitochondria is caused by detaching of mitochondria with ER.

Providing luciferase or target protein level assays in Mfn^{-/-} cells would improve conclusions on effects on miRNA activity.

As per concern raised by the reviewer, we have also provided luciferase assays to check differential miRNA-mediated target protein repression in Mfn2^{-/-} cells on Figure 5B and have mentioned it in the text.

Language should be revised. A few quirks like parallelly and juxta positioning as two words, or wrong verb tenses.

We have tried our best to rectify those problems

References

Bose M, Bhattacharyya SN (2016) Target-dependent biogenesis of cognate microRNAs in human cells. Nature communications 7: 12200

Barman, Bahnisikha, and Suvendra N. Bhattacharyya. "mRNA targeting to endoplasmic reticulum precedes ago protein interaction and microRNA (miRNA)-mediated translation repression in mammalian cells." Journal of Biological Chemistry 290.41 (2015): 24650-24656.

Caudy, Amy A., et al. "Fragile X-related protein and VIG associate with the RNA interference machinery." Genes & development 16.19 (2002): 2491-2496.

Chakrabarty Y, Bhattacharyya SN (2017) Leishmania donovani restricts mitochondrial dynamics to enhance miRNP stability and target RNA repression in host macrophages. Molecular biology of the cell 28: 2091-2105

January 10, 2020

RE: Life Science Alliance Manuscript #LSA-2018-00161-TR-A

Dr. Suvendra N Bhattacharyya
CSIR-Indian Institute of Chemical Biology
Molecular and Human Genetics Division
RNA Biology Research Laboratory, 4 Raja S C Mullick Road
Kolkata, West Bengal 700032
India

Dear Dr. Bhattacharyya,

Thank you for submitting your revised manuscript entitled "Retrograde Trafficking of Argonaute 2 Acts as a Rate-Limiting Step for de novo miRNP Formation". As you will see, the reviewers appreciate the introduced changes and we would thus be happy to publish your paper in Life Science Alliance pending final minor revisions:

- Please address the remaining concerns of reviewer #1
- Please add a callout to figure 7 in the manuscript text
- We only have supplementary figures (not EV figures) - please rename
- Some of the figures have several panels for a single panel descriptor (eg. Fig 1, 2, 6), and it is hard in places to extract what the descriptor-less panels refer to (see for example 1A, 1D (?), 1E (?), 2A...)
- please consider introducing more panel descriptors and to also add more information to the graphs (eg. to the Mander's coefficient graphs where it is currently unclear to which staining they refer to).
- Please make sure that the insets in Fig 6A and S3D really match. I would also recommend to use higher resolution images (if at hand) and to reduce the thickness of the box outline to avoid losing too much information

A. FINAL FILES:

B. MANUSCRIPT ORGANIZATION AND FORMATTING:

Sincerely,

Andrea Leibfried, PhD
Executive Editor
Life Science Alliance
Meyerhofstr. 1

69117 Heidelberg, Germany
t +49 6221 8891 502
e a.leibfried@life-science-alliance.org
www.life-science-alliance.org

Reviewer #1 (Comments to the Authors (Required)):

First paragraph of Introduction, last sentence: "endonucleolytic" should be "exonucleolytic". The sentence also needs English editing.

Figure 5F:

The panel on the left shows increased target mRNA associated with rER. The panel on the right is meant to show the reduced association of miR-122 with rER. However, the level of miR-122 is normalized to the level of the target. As the target is increased, it is not surprising that the miR-122 level is reduced. Another internal control should be used.

Second paragraph of Discussion, first sentence: "extracellular" should be "intracellular"; "export" should be "transport"

Reviewer #2 (Comments to the Authors (Required)):

I appreciate the complexity of the analyses performed and while they will always have challenges for definitive interpretation I think the authors have improved the clarity of their methods and data to allow them to be understood, and their interpretation is supported by the data. I support publication.

Response to the Reviewer1 Suggestions:

First paragraph of Introduction, last sentence: "endonucleolytic" should be "exonucleolytic". The sentence also needs English editing.

This has been altered appropriately.

Figure 5F:

The panel on the left shows increased target mRNA associated with rER. The panel on the right is meant to show the reduced association of miR-122 with rER. However, the level of miR-122 is normalized to the level of the target. As the target is increased, it is not surprising that the miR-122 level is reduced. Another internal control should be used.

The confusion is apparent but we may have failed to communicate the conclusion we made based on that experiment. The target driven mRNA biogenesis, as published earlier (Bose et al. 2016, *Nature Comm.*), suggests there should be an increased formation of *de novo* miRNAs in presence of its substrate mRNA. The reduced miRNA biogenesis in the *in vitro* system observed here with microsomes isolated from Mfn2 negative cells could therefore be due to reduced target mRNA associated with it. The data in panel 5H (old 5F) *left* shows that there was increased target RNA instead. In a miRNA-limiting non-steady state where miRNA biogenesis was only followed, it was not expected that the target RNA should show a reciprocal relation with miRNA content that otherwise holds true, as the reviewer has also pointed out, for steady state *in vivo* situation. Therefore, we rather expect a increased miRNA biogenesis with increased target in the condition we adopted. However, with Mfn2 loss, the situation got reversed and we had detected a reduced miRNA biogenesis in Mfn2 negative context favouring involvement of Mfn2/Mitochondria in the miRNA biogenesis process.

However, following the reviewer's suggestion, we have done normalization against Ago2 content also and in this case we have detected a reduced miRNA biogenesis for per unit of Ago2 protein present as we have observed also in normalization done against target RNA content.

Second paragraph of Discussion, first sentence: "extracellular" should be "intracellular"; "export" should be "transport"

It has been the EV-mediated extracellular miRNA export that was referred there. Therefore no change is required.

Response to the comments by the Editor:

- Please address the remaining concerns of reviewer #1

Please see our reply to the Reviewer1 comments.

- Please add a callout to figure 7 in the manuscript text.

It has been done .

- *We only have supplementary figures (not EV figures) - please rename*

The EV figures have been renamed throughout the text.

- *Some of the figures have several panels for a single panel descriptor (eg. Fig 1, 2, 6), and it is hard in places to extract what the descriptor-less panels refer to (see for example 1A, 1D (?), 1E (?), 2A...)*
- *please consider introducing more panel descriptors and to also add more information to the graphs (eg. to the Mander's coefficient graphs where it is currently unclear to which staining they refer to).*

We have introduced multiple panel descriptors as appropriate in places. The additional information in graph descriptions like in Mander's coefficient have also been inserted.

- *Please make sure that the insets in Fig 6A and S3D really match. I would also recommend to use higher resolution images (if at hand) and to reduce the thickness of the box outline to avoid losing too much information.*

Larger images are used and magnification fold has been indicated as appropriate. The outlined boxes thickness has been reduced and properly placed in both Figures.

January 15, 2020

RE: Life Science Alliance Manuscript #LSA-2018-00161-TRR

Dr. Suvendra N Bhattacharyya
CSIR-Indian Institute of Chemical Biology
Molecular and Human Genetics Division
RNA Biology Research Laboratory, 4 Raja S C Mullick Road
Kolkata, West Bengal 700032
India

Dear Dr. Bhattacharyya,

Thank you for submitting your Research Article entitled "Retrograde Trafficking of Argonaute 2 Acts as a Rate-Limiting Step for de novo miRNP Formation". I appreciate the introduced changes and it is a pleasure to let you know that your manuscript is now accepted for publication in Life Science Alliance. Congratulations on this interesting work.

DISTRIBUTION OF MATERIALS:

Again, congratulations on a very nice paper. I hope you found the review process to be constructive and are pleased with how the manuscript was handled editorially. We look forward to future exciting submissions from your lab.

Sincerely,

Andrea Leibfried, PhD
Executive Editor
Life Science Alliance
Meyershofstr. 1
69117 Heidelberg, Germany
t +49 6221 8891 502
e a.leibfried@life-science-alliance.org
www.life-science-alliance.org